# Distribution-free inference for regression: discrete, continuous, and in between

**Yonghoon Lee**
Department of Statistics
University of Chicago
Chicago, IL 60637
yhoony31@uchicago.edu

**Rina Foygel Barber**
Department of Statistics
University of Chicago
Chicago, IL 60637
rina@uchicago.edu

## Abstract

In data analysis problems where we are not able to rely on distributional assumptions, what types of inference guarantees can still be obtained? Many popular methods, such as holdout methods, cross-validation methods, and conformal prediction, are able to provide distribution-free guarantees for predictive inference, but the problem of providing inference for the underlying regression function (for example, inference on the conditional mean $\mathbb{E}[Y|X]$) is more challenging. In the setting where the features $X$ are continuously distributed, recent work has established that any confidence interval for $\mathbb{E}[Y|X]$ must have non-vanishing width, even as sample size tends to infinity. At the other extreme, if $X$ takes only a small number of possible values, then inference on $\mathbb{E}[Y|X]$ is trivial to achieve. In this work, we study the problem in settings in between these two extremes. We find that there are several distinct regimes in between the finite setting and the continuous setting, where vanishing-width confidence intervals are achievable if and only if the effective support size of the distribution of $X$ is smaller than the square of the sample size.

## 1 Introduction

Consider a regression problem, where our aim is to model the distribution of a response variable $Y \in \mathbb{R}$ based on the information carried by features $X \in \mathcal{X}$. Given training data $(X_1, Y_1), \ldots, (X_n, Y_n)$, we aim to build a fitted model to estimate the conditional distribution of $Y \mid X$, or some summary of this distribution such as the conditional mean or conditional median. In this type of setting, our goals are to simultaneously perform two tasks, estimation and inference—that is, we want to accurately estimate the conditional distribution, and we also want a reliable way of quantifying our uncertainty about this estimate.

To make this concrete, suppose the training data $\{(X_i, Y_i)\}$ are drawn i.i.d. from some unknown distribution $P$ on $\mathbb{R}^d \times \mathbb{R}$, and we want to estimate the true conditional mean, $\mu_P(x) := \mathbb{E}[Y|X=x]$, of this distribution. Given the training data, we construct a fitted regression function $\widehat{\mu} : \mathbb{R}^d \to \mathbb{R}$ using any algorithm, for instance, a parametric method such as least squares or a nonparametric procedure such as a Gaussian kernel method. For many regression algorithms, assuming certain conditions on the underlying distribution $P$ will ensure an accurate estimate of $\mu_P$; however, unless we are able to verify these assumptions, we cannot be confident that the corresponding error rates will indeed lead to a valid confidence interval for $\mu_P$. The goal of *distribution-free inference* is to provide inference guarantees—in this case, confidence intervals for $\mu_P(X_{n+1})$ at a newly observed feature vector $X_{n+1}$—that are valid universally over any underlying distribution $P$.

35th Conference on Neural Information Processing Systems (NeurIPS 2021).

## 1.1 Our contributions

In this work, we study the problem of constructing a confidence interval $\widehat{C}_n(x)$ for $\mu_P(x)$, that satisfies the following property:

**Definition 1.** *An algorithm $\widehat{C}_n$ provides a distribution-free $(1 - \alpha)$-confidence interval for the conditional mean if it holds that*

$$\mathbb{P}_{(X_i, Y_i) \overset{\text{iid}}{\sim} P} \left\{ \mu_P(X_{n+1}) \in \widehat{C}_n(X_{n+1}) \right\} \geq 1 - \alpha \text{ for all distributions } P \text{ on } (X, Y) \in \mathbb{R}^d \times [0, 1].$$

Here the probability is taken with respect to the distribution of both the training data $(X_1, Y_1), \ldots, (X_n, Y_n)$ and the test point $(X_{n+1}, Y_{n+1})$, all drawn i.i.d. from an arbitrary $P$.[1]

Recent work by Vovk et al. [2005], Barber [2020], Gupta et al. [2020] (studying the conditional mean of a binary response $Y$) and by Medarametla and Candès [2021] (studying the conditional median of a real-valued $Y$) proves that distribution-free coverage properties similar to Definition 1 lead to fundamental limits on the accuracy of inference. Writing $P_X$ to denote the marginal distribution of $X$ under $P$, these results show that if $P_X$ is *nonatomic* (meaning that there are no point masses, i.e., $\mathbb{P}_{P_X} \{X = x\} = 0$ for all points $x \in \mathbb{R}^d$), then any distribution-free confidence interval $\widehat{C}_n$ cannot have vanishing length as sample size $n$ tends to infinity, regardless of the smoothness of $P$, or any other "nice" properties of this distribution. Specifically, these works show that if $P_X$ is nonatomic, then $\widehat{C}_n$ must also be a valid predictive interval, i.e., must contain $Y_{n+1}$ itself with probability $\geq 1 - \alpha$. This implies that the length of $\widehat{C}_n$ cannot be vanishing, since $Y_{n+1}$ is inherently noisy. An explicit lower bound on the length is proved in Barber [2020].

Our new results examine the possibility of constructing confidence intervals $\widehat{C}_n$ that are both distribution-free (Definition 1) and have vanishing length, when $P_X$ may be discrete, nonatomic, or a mixture of the two. We find that the hardness of this problem can be characterized by the *effective support size* of $P_X$—essentially, how many points $x \in \mathbb{R}^d$ are needed to capture most of the mass of $P_X$ (for example, if $P_X$ is uniform over $M$ points, then its effective support size is $\leq M$).

Our main theoretical results show that there are two regimes. If the effective support size is $\gg n^2$, then $P_X$ essentially behaves like a nonatomic distribution because in a sample of size $n$, with high probability all the $X$ values are observed at most once; in this regime, we find that the average length of $\widehat{C}_n(X_{n+1})$ is bounded away from zero, i.e., no distribution-free confidence interval can have vanishing length. If instead the effective support size is $\ll n^2$, then it becomes possible for $\widehat{C}_n(X_{n+1})$ to have vanishing length, and in particular, the minimum possible length scales as $\frac{M^{1/4}}{n^{1/2}}$ for effective support size $M$. Interestingly, vanishing length is possible even when $M$ is larger than $n$, meaning that distribution-free inference for $\mathbb{E}[Y|X]$ is possible even if most $X$ values were never observed in the training set.

## 1.2 Additional related work

The problem of distribution-free inference has been studied extensively in the context of *predictive inference*, where the goal is to provide a confidence band for the response value $Y_{n+1}$ given a new feature vector $X_{n+1}$. The prediction problem is fundamentally different from the goal of covering the conditional mean. In particular, by splitting the data and using a holdout set, we can always empirically validate the coverage level of any constructed predictive band. Methods such as conformal prediction (see, e.g., Vovk et al. [2005], Papadopoulos et al. [2002], Lei et al. [2018], Vovk et al. [2018]) or jackknife+ [Barber et al., 2021, Kim et al., 2020] can ensure valid distribution-free predictive inference without the need to split the data set (thus avoiding reducing the sample size).

As mentioned earlier, Vovk et al. [2005], Barber [2020], Gupta et al. [2020], Medarametla and Candès [2021] also study the problem of confidence intervals for the conditional mean or median of $Y|X$, establishing impossibility results on the setting of a nonatomic $P_X$. These results are connected to earlier results on the impossibility of adaptation to smoothness, in the nonparametric inference literature—specifically, if $\mu_P$ is $\beta$-Hölder smooth, then it is possible to build a confidence interval of

---

[1]In this definition and throughout our work, $\widehat{C}_n$ can be either a deterministic or randomized function of the training data; if the construction is randomized then the definition above should be interpreted as computing probability with respect to the distribution of the data and the randomization of the construction.

length $\mathcal{O}(n^{-\frac{\beta}{2\beta+d}})$ if $\beta$ is known (e.g., using $k$-nearest-neighbors with an appropriately chosen $k$), but this cannot be achieved when $\beta$ is unknown (see, e.g., Giné and Nickl [2016, Section 8.3] for an overview of results of this type).

While the results above establish the challenges for distribution-free inference when the features $X$ are nonatomic, at the other extreme we can consider scenarios where $X$ has a discrete distribution. In this setting, the problem of estimating $\mu_P$ is related to the *discrete distribution testing*, where the aim is to test properties of a discrete distribution—for instance, we might wish to test equality of two distributions where we draw samples from each [Chan et al., 2014, Acharya et al., 2014, Diakonikolas and Kane, 2016, Canonne et al., 2015]; to test whether a sample is drawn from a known distribution $P$ or not [Diakonikolas and Kane, 2016, Acharya et al., 2015, Valiant and Valiant, 2017, Diakonikolas et al., 2018], or drawn from any distribution belonging to a class $\mathcal{P}$ or not [Acharya et al., 2015, Canonne et al., 2018]; or to estimate certain characteristics of a distribution such as its entropy or support size [Valiant and Valiant, 2011b,a, Acharya et al., 2014]. The distribution-free confidence intervals we will construct in Section 3 are closely related to methods developed in this literature.

## 2 Main results: lower bound

Before presenting our main result, we begin with several definitions. For any distribution $P_X$ on $X \in \mathbb{R}^d$, we first define the *effective support size* of $P_X$ at tolerance level $\gamma \in [0, 1)$:

$$M_\gamma(P_X) = \min \left\{ |\mathcal{X}| \ : \ \mathcal{X} \subset \mathbb{R}^d \text{ and } \mathbb{P}_{P_X} \{X \in \mathcal{X}\} \geq 1 - \gamma \right\},$$

where $|\mathcal{X}|$ denotes the cardinality of the set $\mathcal{X}$. In particular, if $P_X$ is a distribution supported on $M$ points, then $M_\gamma(P_X) \leq M$ for any $\gamma$. If instead $P_X$ is nonatomic, then $M_\gamma(P_X) = \infty$ for all $\gamma > 0$. We note that, in many practical settings, the effective support size $M_\gamma(P_X)$ may be substantially smaller than the overall support size. For example, if $X \in \mathbb{R}^d$ measures $d$ categorical covariates with $m_j$ possible values for the $j$th covariate, then the support of $P_X$ is potentially as large as $\prod_{j=1}^d m_j$, which will grow extremely rapidly with the dimension $d$ even if each $m_j$ is small; in real data, however, it may be the case that most combinations of covariate values are extremely unlikely, and so the effective support size $M_\gamma(P)$ would be substantially smaller, and might grow more slowly with $d$.

Next, for any distribution $P$ on $(X, Y) \in \mathbb{R}^d \times [0, 1]$, we define

$$\sigma_{P,\beta}^2 = \text{ the } \beta\text{-quantile of } \mathrm{Var}_P(Y|X), \text{ under the distribution } X \sim P_X.$$

With these definitions in place, our first main result establishes a lower bound on the expected length of any distribution-free confidence interval $\widehat{C}_n$. Let Leb denote the Lebesgue measure on $\mathbb{R}$.

**Theorem 1.** *Fix any $\alpha > 0$, and let $\widehat{C}_n$ be a distribution-free $(1-\alpha)$-confidence interval (i.e., satisfying Definition 1). Then for any distribution $P$ on $\mathbb{R}^d \times \mathbb{R}$, for any $\beta > 0$ and $\gamma > \alpha + \beta$,*

$$\mathbb{E}\left[\mathrm{Leb}\left(\widehat{C}_n(X_{n+1})\right)\right] \geq \tfrac{1}{3}\sigma_{P,\beta}^2(\gamma - \alpha - \beta)^{1.5} \cdot \min\left\{ \frac{\left(M_\gamma(P_X)\right)^{1/4}}{n^{1/2}}, 1 \right\},$$

*where the expected value is taken over data points $(X_i, Y_i) \overset{\text{iid}}{\sim} P$, for $i = 1, \dots, n+1$.*

### 2.1 Special cases

To help interpret this result, we now examine its implications in several special cases.

**Uniform discrete features** If $P_X$ is a uniform distribution over $M$ points, then for any $\gamma > 0$ the effective support size is $M_\gamma(P_X) = \lceil (1-\gamma)M \rceil$. Therefore, Theorem 1 implies that for any $P$ with nonatomic marginal $P_X$,

$$\mathbb{E}\left[\mathrm{Leb}\left(\widehat{C}_n(X_{n+1})\right)\right] \geq \tfrac{1}{3}\sigma_{P,\beta}^2(\gamma - \alpha - \beta)^{1.5}(1-\gamma)^{0.25} \cdot \min\left\{ \frac{M^{1/4}}{n^{1/2}}, 1 \right\}$$

for any $\beta \in (0, \gamma - \alpha)$. In particular, we see that $M \gg n^2$ implies a *constant* lower bound on the width of any distribution-free confidence interval, while $M \ll n^2$ allows for the possibility of a *vanishing* width for a distribution-free confidence interval.

**Binary response** If the response $Y$ is known to be binary (i.e., $Y \in \{0, 1\}$), we might relax the requirement of distribution-free coverage to only include distributions of this type, i.e., we require

$$\mathbb{P}_{(X_i, Y_i) \overset{\text{iid}}{\sim} P} \left\{ \mu_P(X_{n+1}) \in \widehat{C}_n(X_{n+1}) \right\} \geq 1 - \alpha \text{ for all distributions } P \text{ on } \mathbb{R}^d \times \{0, 1\}. \quad (1)$$

This condition is strictly weaker than Definition 1, where the coverage property is required to hold for all distributions $P$ on $\mathbb{R}^d \times [0, 1]$, i.e., for a broader class of distributions. However, it turns out that relaxing the requirement does not improve the lower bound. Specifically, if we have an algorithm to construct a confidence interval $\widehat{C}_n$ satisfying (1), then we can easily convert $\widehat{C}_n$ into a method that does satisfy Definition 1. Given data $(X_1, Y_1), \ldots, (X_n, Y_n)$, for each $i = 1, \ldots, n$ draw a binary response $\tilde{Y}_i \sim \text{Bernoulli}(Y_i)$. Then we clearly have $n$ i.i.d. draws from a distribution on $(X, \tilde{Y}) \in \mathbb{R}^d \times \{0, 1\}$, where $\mathbb{E}[\tilde{Y} \mid X] = \mathbb{E}[Y \mid X] = \mu_P(X)$. After running our algorithm to construct $\widehat{C}_n$ on the new data $(X_1, \tilde{Y}_1), \ldots, (X_n, \tilde{Y}_n)$, the binary distribution-free coverage property (1) satisfied by $\widehat{C}_n$ ensures that this modified procedure satisfies Definition 1.

To summarize, then, we see that the problem of distribution-free coverage is equally hard for the binary response case ($Y \in \{0, 1\}$) as for the more general bounded response case ($Y \in [0, 1]$).

**Nonatomic features** We now consider the setting where the marginal distribution of $X$ is nonatomic, i.e., $\mathbb{P}_{P_X} \{X = x\} = 0$ for all $x$. (In particular, this includes the continuous case, where $X$ has a continuous distribution on $\mathbb{R}^d$.) In this case, for any $\gamma > 0$ the effective support size is $M_\gamma(P_X) = \infty$. Therefore, Theorem 1 implies that for any $P$ with nonatomic marginal $P_X$, for any $\beta \in (0, 1 - \alpha)$,

$$\mathbb{E}\left[ \text{Leb}\left( \widehat{C}_n(X_{n+1}) \right) \right] \geq \tfrac{1}{3} \sigma_{P,\beta}^2 (1 - \alpha - \beta)^{1.5}.$$

In particular, this lower bound does not depend on $n$, and so the width of any distribution-free confidence interval is non-vanishing even for arbitrarily large sample size $n$ (as long as $\sigma_{P,\beta}^2 > 0$).

In case of a binary response, where $P$ is a distribution on $\mathbb{R}^d \times \{0, 1\}$ with nonatomic marginal distribution $P_X$, Barber [2020] establishes that any distribution-free confidence interval for $\mu$ must satisfy a lower bound that is a function only of $P$ and does not depend on $n$ (and, in particular, does not vanish as $n \to \infty$). In this sense, our new result can be viewed as a generalization of this work, since the nonvanishing minimum length for nonatomic $P_X$ is a consequence of our result.

## 2.2 Adding knowledge of $P_X$

One way we might try to weaken the notion of distribution-free coverage would be to allow assumptions about the marginal distribution $P_X$, while remaining assumption-free for the function $\mu_P$ determining the conditional mean. In other words, we might weaken Definition 1 to require coverage over all distributions $P$ for which $P_X = P_X^*$, for a known $P_X^*$ (or, all $P$ for which $P_X$ satisfies some assumed property). Interestingly, the lower bound in Theorem 1 remains the same even under this milder definition of validity—we will see in the proof that knowledge of $P_X$ does not affect the lower bound, since the argument relies only on our uncertainty about the conditional distribution of $Y|X$.

## 2.3 Bounded or unbounded?

The lower bound established in Theorem 1 assumes distribution-free coverage for distributions with a bounded response $Y$—that is, Definition 1 requires coverage to hold for distributions where the response $Y$ is supported on $[0, 1]$ (although no other assumptions are placed on $P$). Would it be possible for us to instead consider the general case, where $P$ is an unknown distribution on $\mathbb{R}^d \times \mathbb{R}$? The following result shows that this more general question is not meaningful:

**Proposition 1.** *Suppose an algorithm $\widehat{C}_n$ satisfies*

$$\mathbb{P}_{(X_i, Y_i) \overset{\text{iid}}{\sim} P} \left\{ \mu_P(X_{n+1}) \in \widehat{C}_n(X_{n+1}) \right\} \geq 1 - \alpha \text{ for all distributions } P \text{ on } \mathbb{R}^d \times \mathbb{R}.$$

*Then for all distributions $P$, for all $y \in \mathbb{R}$ it holds that*

$$\mathbb{P}_{(X_i, Y_i) \overset{\text{iid}}{\sim} P} \left\{ y \in \widehat{C}_n(X_{n+1})) \right\} \geq 1 - \alpha.$$

This means that if we require $\widehat{C}_n$ to have distribution-free coverage over distributions with *unbounded* response, then inevitably, *every point in the real line* is contained in the resulting confidence interval a substantial portion of the time. (In particular, $\widehat{C}_n(X_{n+1})$ will of course have infinite expected width.) Clearly an unbounded $Y$ cannot result in any meaningful distribution-free inference, and for this reason we therefore restrict our attention to the setting where the response $Y$ takes values in $[0,1]$ (of course, these results can easily generalize to $Y \in [a,b]$ for any known $a < b$).

## 3    Main results: upper bound

We next construct an algorithm that, for certain "nice" distributions $P$, can achieve a confidence interval length that matches the rate of the lower bound. Our procedure requires two main ingredients as input:

1. A hypothesized ordered support set $\{x^{(1)}, x^{(2)}, \dots\} \subset \mathbb{R}^d$ for the marginal $P_X$, and
2. A hypothesized mean function $\mu : \mathbb{R}^d \to [0,1]$.

One possible way of obtaining these inputs would be to use data splitting, where one portion of our data (combined with prior knowledge if available) is used to construct a hypothesized support set and mean function, and the second portion of the data is then used for constructing the confidence interval (note that the sample size $n$ in our construction below refers to the size of this second part of the data, e.g., half of the total available sample size). Any algorithm can be applied for estimating $\mu$, for example, logistic regression, nearest neighbors regression, or a neural network.

We emphasize that the coverage guarantee provided by our method does not rely in any way on the accuracy of these initial guesses—the constructed confidence interval will satisfy distribution-free validity (Definition 1) even if these initial parameters are chosen in a completely uninformed way. In particular, while the algorithm that fits $\mu$ might be able to guarantee accuracy of $\mu$ under some assumptions placed on $P$, the validity of our inference procedure does not rely on these assumptions. However, the length of the resulting confidence interval will be affected, since high accuracy in these initial guesses can be expected to result in a shorter confidence interval. In particular, the hypothesized support set $\{x^{(1)}, x^{(2)}, \dots\}$ should aim to list the highest-probability values of $X$ early in the list, while the hypothesized mean function $\mu$ should aim to be as close to the true conditional mean $\mu_P$ as possible. (Our theoretical results below will make these goals more precise.)

Given the hypothesized support and hypothesized mean function, to run our algorithm, we first choose parameters $\gamma, \delta > 0$ satisfying $\gamma + \delta < \alpha$, and then compute the following steps.

- **Step 1: estimate the effective support size.** First, we compute an upper bound on the support size needed to capture $1 - \gamma$ of the probability under $P_X$,

$$\widehat{M}_\gamma = \min\left\{ m : \sum_{i=1}^n \mathbb{1}\left\{ X_i \in \{x^{(1)}, \dots, x^{(m)}\} \right\} \geq (1-\gamma)n + \sqrt{\frac{n \log(2/\delta)}{2}} \right\},$$

or $\widehat{M}_\gamma = \infty$ if there is no $m$ that satisfies the inequality. Applying the Hoeffding inequality to the $\mathrm{Binom}(n, \gamma)$ distribution, we see that $\mathbb{P}\left\{ \widehat{M}_\gamma \geq M_\gamma^*(P_X) \right\} \geq 1 - \delta/2$, where

$$M_\gamma^*(P_X) = \min\left\{ m : \mathbb{P}_{P_X}\left\{ X \in \{x^{(1)}, \dots, x^{(m)}\} \right\} \geq 1 - \gamma \right\}. \tag{2}$$

(Note that $M_\gamma^*(P_X) \geq M_\gamma(P_X)$ by definition.)

- **Step 2: estimate error at each repeated $X$ value.** Next, for each $m = 1, 2, \dots$, let $n_m = \sum_{i=1}^n \mathbb{1}\left\{ X_i = x^{(m)} \right\}$ denote the number of times $x^{(m)}$ was observed, and let

$$N_{\geq 2} = \sum_{m \geq 1} \mathbb{1}\left\{ n_m \geq 2 \right\} \tag{3}$$

be the number of $X$ values observed at least twice. For each $m$ with $n_m \geq 2$, let $\bar{y}_m = \frac{1}{n_m} \sum_{i=1}^n Y_i \cdot \mathbb{1}\left\{ X_i = x^{(m)} \right\}$ and $s_m^2 = \frac{1}{n_m - 1} \sum_{i=1}^n (Y_i - \bar{y}_m)^2 \cdot \mathbb{1}\left\{ X_i = x^{(m)} \right\}$ be the sample mean and sample variance of the corresponding $Y$ values. Define

$$Z = \sum_{\substack{m=1,2,\dots \\ \text{s.t. } n_m \geq 2}} (n_m - 1) \cdot \left( (\bar{y}_m - \mu(x^{(m)}))^2 - n_m^{-1} s_m^2 \right). \tag{4}$$

This construction is inspired by analogous statistics appearing in the literature for testing properties of discrete distributions—for instance, the work of Chan et al. [2014]. To see the intuition behind this construction, we observe that $\{Y_i : X_i = x^{(m)}\}$ is a collection of i.i.d. observations with mean $\mu_P(x^{(m)})$. Therefore, conditional on $n_m$ (with $n_m \geq 2$),

$$\mathbb{E}\left[\bar{y}_m\right] = \mu_P(x^{(m)}) \text{ and } \mathrm{Var}\left(\bar{y}_m\right) = n_m^{-1}\mathbb{E}\left[s_m^2\right],$$

and therefore $\mathbb{E}\left[(\bar{y}_m - \mu(x^{(m)}))^2 - n_m^{-1}s_m^2\right] = (\mu(x^{(m)}) - \mu_P(x^{(m)}))^2$ is an estimate of our error at this $X$ value.

- **Step 3: construct the confidence interval.** Finally, we define our confidence interval. Let

$$\widehat{\Delta} = \sqrt{\frac{2\widehat{M}_\gamma + n}{n(n-1)}} \cdot \sqrt{4Z_+ + 8\sqrt{N_{\geq 2}/\delta} + 24/\delta},$$

where $Z_+$ denotes $\max\{Z, 0\}$. Then for each $x \in \mathbb{R}^d$, we define

$$\widehat{C}_n(x) = \left[\max\left\{0, \mu(x) - \frac{\widehat{\Delta}}{\alpha - \delta - \gamma}\right\}, \min\left\{1, \mu(x) + \frac{\widehat{\Delta}}{\alpha - \delta - \gamma}\right\}\right]. \quad (5)$$

We now verify that this construction yields a valid distribution-free confidence interval.

**Theorem 2.** *The confidence interval constructed in* (5) *is a distribution-free* $(1 - \alpha)$-*confidence interval (i.e.,* $\widehat{C}_n$ *satisfies Definition 1).*

Next, we will see how this construction is able to match the rate of the lower bound established in Theorem 1—specifically, in a scenario where the hypothesized support set and mean function are "chosen well", i.e., are a good approximation to the true distribution $P$. For simplicity, we only consider the case where the marginal $P_X$ is approximately uniform over some finite subset of the hypothesized support, and the hypothesized function $\mu$ has uniformly bounded error.

**Theorem 3.** *Suppose the distribution $P$ on $(X, Y) \in \mathbb{R}^d \times \mathbb{R}$ has marginal $P_X$ that is supported on $\{x^{(1)}, \ldots, x^{(M)}\}$ and satisfies $\mathbb{P}_{P_X}\left\{X = x^{(m)}\right\} \leq \eta/M$ for all $m$, and suppose that $P$ has conditional mean $\mu_P : \mathbb{R}^d \to \mathbb{R}$ that satisfies $\mathbb{E}_{P_X}\left[(\mu_P(X) - \mu(X))^2\right] \leq \mathrm{err}_\mu^2$. Then the confidence interval constructed in* (5) *satisfies*

$$\mathbb{E}\left[\mathsf{Leb}(\widehat{C}_n(X_{n+1}))\right] \leq c\left(\mathrm{err}_\mu + \frac{M^{1/4}}{n^{1/2}}\right),$$

*where $c$ depends only on the parameters $\alpha$, $\delta$, $\gamma$, $\eta$.*

To see some concrete examples of where this upper bound might be small, suppose that $\mu$ is constructed via data splitting (i.e., our initial data set has sample size $2n$, and we use $n$ data points to train $\mu$ and then the remaining $n$ to construct the confidence interval). If $\mu$ is constructed via logistic regression, and the distribution $P$ follows this model, then under standard conditions on $P_X$ we would have $\mathrm{err}_\mu = \mathcal{O}(\sqrt{d/n})$; in a $k$-sparse regression setting where we use logistic lasso we might instead obtain $\mathrm{err}_\mu = \mathcal{O}(\sqrt{k\log(d)/n})$ [Negahban et al., 2012]. If instead $\mu$ is constructed via $k$-nearest neighbors, if $x \mapsto \mu_P(x)$ is $\beta$-Hölder smooth (and $k$ is chosen appropriately), then as mentioned earlier we have $\mathrm{err}_\mu = \mathcal{O}(n^{-\beta/(\beta+d)})$ [Györfi et al., 2002, Giné and Nickl, 2016].

## 4 Discussion

Our main result, Theorem 1, shows that the problem of constructing distribution-free confidence intervals for a conditional mean has hardness characterized by the effective support size $M_\gamma(P_X)$ of the feature distribution; distribution-free confidence intervals may have vanishing length if the sample size is at least as large as the square root of the effective support size, but must have length bounded away from zero if the sample size is smaller. The rate of the lower bound on length, scaling as $\min\{\frac{M_\gamma(P_X))^{1/4}}{n^{1/2}}, 1\}$, is achievable in certain settings—Theorems 2 and 3 establish that distribution-free confidence intervals may achieve this length if we have a good hypothesis $\mu$ for $\mu_P$. Of course, the specific construction used for these matching bounds may not be optimal—both in

terms of constant factors that may inflate its length, and in terms of the range of settings in which it is able (up to constants) to match the lower bound. Improving this construction to provide a practical and accurate algorithm is an important question for future work.

One counterintuitive implication of our result is that a meaningful distribution-free inference can be achieved even in the case $M_\gamma(P_X) \gg n$, where with high probability, the new observation $X_{n+1}$ is a value that was never observed in the training set. The reason inference is possible in this regime is that the repeated $X$ values in the training set provide some information we need to construct a meaningful confidence interval, and since the set of $X$ values that are repeated is random, this leads to a coverage guarantee (recall that these repeated $X$ values were central to the construction of our confidence interval in Section 3). An interesting possible application of this finding is for distribution-free calibration, where the aim is to cover within-bin averages of the form $\mu_b = \mathbb{E}[Y \mid X \in \mathcal{X}_b]$ where $\mathbb{R}^d = \cup_{b=1,\ldots,B} \mathcal{X}_b$ is a partition into bins. Gupta et al. [2020] study this problem in the distribution-free setting, and develop methods for guaranteeing coverage of *each* $\mu_b$ when the number of bins satisfies $B \ll n$; in contrast, the methods studied in our present work suggest that we may be able to cover $\mu_b$ *on average* over all bins $b$ in the regime $n \ll B \ll n^2$.

Generally, all inference methods must inherently involve a tradeoff between the strength of the guarantees, and the precision of the resulting answers. In this present work, we consider a universally strong guarantee (i.e., coverage of the conditional mean for *all* distributions $P$), which results in precise inference (i.e., vanishing-length confidence intervals) for only *some* distributions $P$, namely, those with effective support size $\ll n^2$. This tradeoff may not be desirable in practice, since in an applied setting we might instead prefer to relax the required coverage properties for more challenging distributions $P$ in order to allow for more precise answers. In practice, we may be satisfied with a validity condition that yields weaker guarantees in a nonatomic setting, but still yields the stronger coverage guarantee in the achievable regime where $M_\gamma(P_X) \ll n^2$. In future work, we aim to study whether this more adaptive type of validity definition, which is weaker than distribution-free coverage, may enable us to build confidence intervals that have vanishing length even in the nonatomic setting.

## A  Proofs

In this section we will prove Proposition 1 and Theorem 1. The proofs of Theorems 2 and 3, as well as of all the supporting lemmas appearing in this section, are given in the Supplementary Materials.

### A.1  Proof of Proposition 1

To prove this proposition, we will consider replacing $P$ with a distribution that places vanishing probability on some extremely large value.[2] Fix any distribution $P$, and any $y \in \mathbb{R}$. For any fixed $\epsilon > 0$, define a new distribution $Q$ as follows:

$$\text{Draw } X \sim P_X, \text{ then draw } Y|X \sim (1-\epsilon)P_{Y|X} + \epsilon\delta_{\epsilon^{-1}y-(\epsilon^{-1}-1)\mu_P(X)},$$

where $P_{Y|X}$ is the conditional distribution of $Y|X$ under $P$, and $\delta_t$ denotes the point mass at $t$. Then we can trivially calculate that $d_{\text{TV}}(P^n \times P_X, Q^n \times Q_X) \leq n\epsilon$. Therefore,

$$\mathbb{P}_{P^n \times P_X}\left\{y \in \widehat{C}_n(X_{n+1})\right\} \geq \mathbb{P}_{Q^n \times Q_X}\left\{y \in \widehat{C}_n(X_{n+1})\right\} - n\epsilon.$$

On the other hand, the distribution $Q$ has conditional mean

$$\mu_Q(x) = (1-\epsilon)\mu_P(x) + \epsilon\left(\epsilon^{-1}y - (\epsilon^{-1}-1)\mu_P(x)\right) = y,$$

and so the conditional mean $\mu_Q(X_{n+1})$ is equal to $y$ almost surely. Therefore,

$$\mathbb{P}_{Q^n \times Q_X}\left\{y \in \widehat{C}_n(X_{n+1})\right\} = \mathbb{P}_{Q^n \times Q_X}\left\{\mu_Q(X_{n+1}) \in \widehat{C}_n(X_{n+1})\right\} \geq 1 - \alpha,$$

where the last step holds since $\widehat{C}_n$ must satisfy distribution-free coverage and, therefore, must satisfy coverage with respect to $Q$. Since $\epsilon > 0$ is arbitrarily small, this completes the proof.

---

[2]Similar constructions are used in many related results in the literature—e.g., Lei and Wasserman [2014, Lemma 1] proves an analogous infinite-width result for the problem of prediction intervals required to be valid conditional on $X_{n+1}$, while here we are interested in confidence intervals but only require marginal validity.

## A.2  Proof of Theorem 1

To prove the theorem, we will need several supporting lemmas:

**Lemma 1.** *Let $Q$ be any distribution on $[0,1]$ with variance $\sigma^2$. Then we can write $Q$ as a mixture of two distributions $Q_0, Q_1$ on $[0,1]$ such that*

$$Q = 0.5Q_0 + 0.5Q_1 \text{ and } \mathbb{E}_{Q_1}[X] - \mathbb{E}_{Q_0}[X] \geq 2\sigma^2.$$

**Lemma 2.** *Let $P_X$ be any distribution on $\mathbb{R}^d$, and let $\mathbb{R}^d = \mathcal{X}_1 \cup \mathcal{X}_2 \cup \ldots$ be a fixed partition. Define a distribution $P_0$ on $(X, Z) \in \mathbb{R}^d \times \{0,1\}$ as:*

Draw $X \sim P_X$, and draw $Z \sim \text{Bernoulli}(0.5)$, independently from $X$.

*For any fixed sequence $a = (a_1, a_2, \dots)$ of signs $a_1, a_2, \dots \in \{\pm 1\}$, and any fixed $\epsilon_1, \epsilon_2, \dots \in [0, 0.5]$, define a distribution $P_a$ on on $(X, Z) \in \mathbb{R}^d \times \{0,1\}$ as:*

Draw $X \sim P_X$, and conditional on $X$, draw $Z | X \in \mathcal{X}_m \sim \text{Bernoulli}(0.5 + a_m \cdot \epsilon_m)$.

*Finally define $\tilde{P}_0 = (P_0)^n$ (i.e., $n$ i.i.d. draws from $P_0$), and define a mixture distribution $\tilde{P}_1$ on $(X_1, Z_1), \dots, (X_n, Z_n)$ as*

Draw $A_1, A_2, \dots \overset{\text{iid}}{\sim} \text{Unif}\{\pm 1\}$, then conditional on $A = (A_1, A_2, \dots)$, draw $(X_i, Z_i) \overset{\text{iid}}{\sim} P_A$.

*Then*

$$d_{\text{TV}}(\tilde{P}_0, \tilde{P}_1) \leq 2n \sqrt{\sum_{m \geq 1} \epsilon_m^4 \cdot \mathbb{P}_{P_X}\{X \in \mathcal{X}_m\}^2}.$$

We are now ready to prove the theorem. Define $\mathcal{X}_1 = \{x \in \mathbb{R}^d : \mathbb{P}_{P_X}\{X = x\} > \frac{1}{M_\gamma(P_X)}\}$. We must have $|\mathcal{X}_1| < M_\gamma(P_X)$ since $P_X$ is a probability measure, and therefore $\mathbb{P}_{P_X}\{X \in \mathcal{X}_1\} < 1 - \gamma$ by definition of $M_\gamma(P_X)$. On the set $\mathbb{R}^d \backslash \mathcal{X}_1$, any point masses of the distribution $P_X$ must each have probability $\leq 1/M_\gamma(P_X)$, by definition of $\mathcal{X}_1$; $P_X$ may also have a nonatomic component. Applying Dudley et al. [2011, Proposition A.1], we can partition $\mathbb{R}^d \backslash \mathcal{X}_1$ into countably many sets, $\mathcal{X}_2 \cup \mathcal{X}_3 \cup \dots$, such that $\mathbb{P}_{P_X}\{X \in \mathcal{X}_m\} \leq 1/M_\gamma(P_X)$ for all $m \geq 2$. Define $p_m = \mathbb{P}_{P_X}\{X \in \mathcal{X}_m\}$.

For each $x$ in the support of $P_X$, let $P_{Y|X=x}$ denote the conditional distribution of $Y$ given $X = x$. By Lemma 1, we can construct distributions $P^1_{Y|X=x}$ and $P^0_{Y|X=x}$ such that

$$P_{Y|X=x} = 0.5P^1_{Y|X=x} + 0.5P^0_{Y|X=x} \text{ and } \mathbb{E}_{P^1_{Y|X=x}}[Y] - \mathbb{E}_{P^0_{Y|X=x}}[Y] \geq 2\sigma_P^2(x),$$

where $\sigma_P^2(x) = \text{Var}(Y \mid X = x)$ is the variance of $P_{Y|X=x}$. Next fix $\epsilon_1 = 0$ and $\epsilon_2 = \epsilon_3 = \dots = \epsilon$ for some $\epsilon \in (0, 0.5]$. For any vector $a = (a_1, a_2, \dots)$ of signs $a_1, a_2, \dots \in \{\pm 1\}$, define the distribution $P_a$ over $(X, Y)$ as follows:

- Draw $X \sim P_X$, i.e., the same as the marginal distribution of $X$ under $P$.
- Conditional on $X = x \in \mathcal{X}_m$ for any $m \geq 1$, draw $Y$ as

$$Y \mid X = x \sim (0.5 + a_m \epsilon) \cdot P^1_{Y|X=x} + (0.5 - a_m \epsilon) \cdot P^0_{Y|X=x}.$$

Thus, $P_a$ differs from $P$ in that, conditional on $X \in \mathcal{X}_m$ for any $m \geq 2$, the distribution of $Y$ is perturbed to be slightly more likely (if $a_m = +1$) or slightly less likely (if $a_m = -1$) to be drawn from $P^1_{Y|X}$ rather than $P^0_{Y|X}$. Finally, we define a mixture distribution $P_{\text{mix}}$ on $(X_1, Y_1), \dots, (X_n, Y_n)$ as:

Draw $A_1, A_2, \dots \overset{\text{iid}}{\sim} \text{Unif}\{\pm 1\}$, then conditional on $A = (A_1, A_2, \dots)$, draw $(X_i, Y_i) \overset{\text{iid}}{\sim} P_A$.

In the Supplementary Materials, we verify that we can apply Lemma 2 to obtain

$$d_{\text{TV}}(P_{\text{mix}}, P^n) \leq 2n \sqrt{\sum_{m \geq 1} \epsilon_m^4 p_m^2} = 2n \sqrt{\sum_{m \geq 2} \epsilon^4 p_m^2} \leq \frac{2\epsilon^2 n}{\sqrt{M_\gamma(P_X)}}, \quad (6)$$

where the last step holds since $p_m \leq 1/M_\gamma(P_X)$ for all $m \geq 2$, by definition.

The remainder of the proof will center on the fact that, if $\epsilon$ is chosen to make the total variation distance between $P^n$ and $P_{\text{mix}}$ sufficiently small, then it is impossible to distinguish between data drawn from $P^n$ or from $P_A^n$ for a random $A$ (i.e., from $P_{\text{mix}}$); since the conditional mean of $Y|X$ differs by $\mathcal{O}(\epsilon)$ between $P$ and $P_A$, this means that our confidence interval for $\mu_P$ will need to have width at least $\mathcal{O}(\epsilon)$. For any $P_a$, since $\widehat{C}_n$ satisfies distribution-free coverage, we have

$$\mathbb{P}_{(P_a)^n \times P_X} \left\{ \mu_{P_a}(X_{n+1}) \in \widehat{C}_n(X_{n+1}) \right\} \geq 1 - \alpha.$$

We can also calculate, for each $m \geq 2$ and each $x \in \mathcal{X}_m$ that lies in the support of $P_X$,

$$\mu_{P_a}(x) = (0.5 + a_m \epsilon)\mathbb{E}_{P_{Y|X=x}^1}[Y] + (0.5 - a_m \epsilon)\mathbb{E}_{P_{Y|X=x}^0}[Y]$$

$$= 0.5 \left( \mathbb{E}_{P_{Y|X=x}^1}[Y] + \mathbb{E}_{P_{Y|X=x}^0}[Y] \right) + a_m \epsilon \left( \mathbb{E}_{P_{Y|X=x}^1}[Y] - \mathbb{E}_{P_{Y|X=x}^0}[Y] \right) = \mu_P(x) + a_m \epsilon \Delta(x),$$

where we write $\Delta(x) = \left( \mathbb{E}_{P_{Y|X=x}^1}[Y] - \mathbb{E}_{P_{Y|X=x}^0}[Y] \right)$. In particular, if $X_{n+1} \notin \mathcal{X}_1$, then $\mu_{P_a}(X_{n+1}) \in \widehat{C}_n(X_{n+1})$ implies $\{\mu_P(X_{n+1}) \pm \epsilon \Delta(X_{n+1})\} \cap \widehat{C}_n(X_{n+1}) \neq \emptyset$. Therefore,

$$\mathbb{P}_{(P_a)^n \times P_X} \left\{ \{\mu_P(X_{n+1}) \pm \epsilon \Delta(X_{n+1})\} \cap \widehat{C}_n(X_{n+1}) \neq \emptyset \right\}$$

$$\geq \mathbb{P}_{(P_a)^n \times P_X} \left\{ \mu_{P_a}(X_{n+1}) \in \widehat{C}_n(X_{n+1}) \right\} - \mathbb{P}_{(P_a)^n \times P_X} \{X_{n+1} \in \mathcal{X}_1\} \geq \gamma - \alpha,$$

since $\mathbb{P}_{P_X} \{X \in \mathcal{X}_1\} < 1 - \gamma$. Since this bound holds for all $a$, and since $P_{\text{mix}}$ is a mixture of distributions $(P_a)^n$, then

$$\mathbb{P}_{P_{\text{mix}} \times P_X} \left\{ \{\mu_P(X_{n+1}) \pm \epsilon \Delta(X_{n+1})\} \cap \widehat{C}_n(X_{n+1}) \neq \emptyset \right\} \geq \gamma - \alpha.$$

By our total variation bound (6), therefore,

$$\mathbb{P}_{P^n \times P_X} \left\{ \{\mu_P(X_{n+1}) \pm \epsilon \Delta(X_{n+1})\} \cap \widehat{C}_n(X_{n+1}) \neq \emptyset \right\} \geq \gamma - \alpha - \frac{2\epsilon^2 n}{\sqrt{M_\gamma(P_X)}}. \tag{7}$$

Now fix some $\epsilon_0 \in [0, 0.5]$. We calculate

$$\text{Leb}\left( \widehat{C}_n(X_{n+1}) \right) = \int_{t \in \mathbb{R}} \mathbb{1}\left\{ t \in \widehat{C}_n(X_{n+1}) \right\} \, dt \geq \int_{t \geq 0} \mathbb{1}\left\{ \{\mu_P(X_{n+1}) \pm t\} \cap \widehat{C}_n(X_{n+1}) \neq \emptyset \right\} \, dt$$

$$\geq \int_{t=0}^{\epsilon_0 \Delta(X_{n+1})} \mathbb{1}\left\{ \{\mu_P(X_{n+1}) \pm t\} \cap \widehat{C}_n(X_{n+1}) \neq \emptyset \right\} \, dt$$

$$= \int_{\epsilon=0}^{\epsilon_0} \mathbb{1}\left\{ \{\mu_P(X_{n+1}) \pm \epsilon \Delta(X_{n+1})\} \cap \widehat{C}_n(X_{n+1}) \neq \emptyset \right\} \cdot \Delta(X_{n+1}) \, d\epsilon$$

$$\geq 2\sigma_{P,\beta}^2 \int_{\epsilon=0}^{\epsilon_0} \mathbb{1}\left\{ \sigma_P^2(X_{n+1}) \geq \sigma_{P,\beta}^2 \text{ and } \{\mu_P(X_{n+1}) \pm \epsilon \Delta(X_{n+1})\} \cap \widehat{C}_n(X_{n+1}) \neq \emptyset \right\} \, d\epsilon,$$

where the last step holds since $\Delta(X_{n+1}) \geq 2\sigma_P^2(X_{n+1})$ by Lemma 1. Applying (7), and since $\mathbb{P}\left\{ \sigma_P^2(X_{n+1}) \geq \sigma_{P,\beta}^2 \right\} \geq 1 - \beta$ by definition of $\sigma_{P,\beta}^2$, we have

$$\mathbb{E}_{P^n \times P_X} \left[ \text{Leb}\left( \widehat{C}_n(X_{n+1}) \right) \right] \geq 2\sigma_{P,\beta}^2 \int_{\epsilon=0}^{\epsilon_0} \left( \gamma - \alpha - \frac{2\epsilon^2 n}{\sqrt{M_\gamma(P_X)}} \right) - \beta \, d\epsilon$$

$$= 2\sigma_{P,\beta}^2 \left[ \epsilon_0(\gamma - \alpha - \beta) - \frac{2\epsilon_0^3 n}{3\sqrt{M_\gamma(P_X)}} \right].$$

Finally, choosing $\epsilon_0 = \min\left\{ \left( \frac{(\gamma - \alpha - \beta)\sqrt{M_\gamma(P_X)}}{2n} \right)^{1/2}, 0.5 \right\}$ yields the desired lower bound.

## Acknowledgments and Disclosure of Funding

R.F.B. was partially supported by the National Science Foundation via grants DMS-1654076 and DMS-2023109, and by the Office of Naval Research via grant N00014-20-1-2337. The authors thank John Lafferty for helpful discussions that inspired this work.

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
