# Supplementary materials for:
# Distribution-free inference for regression: discrete, continuous, and in between

**Yonghoon Lee**
Department of Statistics
University of Chicago
Chicago, IL 60637
yhoony31@uchicago.edu

**Rina Foygel Barber**
Department of Statistics
University of Chicago
Chicago, IL 60637
rina@uchicago.edu

## B  Additional details for proof of Theorem 1

### B.1  Details for (6)

To compare $P$ and $P_a$, we can equivalently characterize these distributions as follows:

- Draw $X \sim P_X$.
- Conditional on $X$, draw $Z \mid X \in \mathcal{X}_m \sim \text{Bernoulli}(0.5)$ (for the distribution $P$, or for the distribution $P_a$ if $m = 1$), or $Z \mid X \in \mathcal{X}_m \sim \text{Bernoulli}(0.5 + a_m\epsilon)$ (for the distribution $P_a$ if $m \geq 2$).
- Conditional on $X, Z$ draw $Y$ as

$$Y \mid X = x, Z = z \sim P_{Y|X=x}^z.$$

Define $\tilde{P}$ as the distribution over $(X, Y, Z)$ induced by $P$, and $\tilde{P}_a$ as the distribution over $(X, Y, Z)$ induced by $P_a$. Then the marginal distribution of $(X, Y)$ under $\tilde{P}$ and under $\tilde{P}_a$ is given by $P$ and by $P_a$, respectively.

Now consider comparing two distributions on triples $(X_1, Z_1, Y_1), \ldots, (X_n, Z_n, Y_n)$. We will compare $\tilde{P}^n$ versus the mixture distribution $\tilde{P}_{\text{mix}}$ defined as follows:

- Draw $A_1, A_2, \ldots \overset{\text{iid}}{\sim} \text{Unif}\{\pm 1\}$.
- Conditional on $A_1, A_2, \ldots$, draw $(X_1, Y_1, Z_1), \ldots, (X_n, Y_n, Z_n) \overset{\text{iid}}{\sim} \tilde{P}_A$.

Since in our characterization above, the distribution of $Y_1, \ldots, Y_n$ conditional on $X_1, \ldots, X_n$ and on $Z_1, \ldots, Z_n$ is the same for both, the only difference lies in the conditional distribution of $Z_1, \ldots, Z_n$ given $X_1, \ldots, X_n$. Therefore, we can apply Lemma 2 with $\epsilon_1 = 0$ and $\epsilon_2 = \epsilon_3 = \cdots = \epsilon$ to obtain

$$d_{\text{TV}}\left(\tilde{P}_{\text{mix}}, \tilde{P}^n\right) \leq 2n\sqrt{\sum_{m \geq 2} \epsilon^4 p_m^2}.$$

Now let $P_{\text{mix}}$ be the marginal distribution of $(X_1, Y_1), \ldots, (X_n, Y_n)$ under $\tilde{P}_{\text{mix}}$. Noting that $P^n$ is the marginal distribution of $(X_1, Y_1), \ldots, (X_n, Y_n)$ under $\tilde{P}^n$, we therefore have

$$d_{\text{TV}}(P_{\text{mix}}, P^n) \leq d_{\text{TV}}\left(\tilde{P}_{\text{mix}}, \tilde{P}^n\right) \leq 2n\sqrt{\sum_{m \geq 2} \epsilon^4 p_m^2}.$$

35th Conference on Neural Information Processing Systems (NeurIPS 2021).

## C Proof of Theorem 2

First, define $p_m = \mathbb{P}_{P_X}\{X = x^{(m)}\}$. The following lemma establishes some results on its support, expected value, and concentration properties of $Z$:

**Lemma C.1.** *For $Z$ and $N_{\geq 2}$ defined as in (4) and (3), the following holds:*

$$\mathbb{E}\left[Z\right] = \sum_{m=1}^{\infty} (\mu(x^{(m)}) - \mu_P(x^{(m)}))^2 \cdot \left(np_m - 1 + (1 - p_m)^n\right),$$

$$\mathbb{E}\left[Z \mid X_1, \dots, X_n\right] = \sum_{m=1}^{\infty} (n_m - 1)_+ \cdot \left(\mu(x^{(m)}) - \mu_P(x^{(m)})\right)^2,$$

$$\operatorname{Var}\left(\mathbb{E}\left[Z \mid X_1, \dots, X_n\right]\right) \leq 2\mathbb{E}\left[Z\right],$$

$$\operatorname{Var}\left(Z \mid X_1, \dots, X_n\right) \leq N_{\geq 2} + 2\mathbb{E}\left[Z \mid X_1, \dots, X_n\right].$$

In particular, the first part of the lemma will allow us to use $\mathbb{E}\left[Z\right]$ to bound the error in $\mu$—here the calculations are similar to those in Chan et al. [2014] for the setting of testing discrete distributions. Recalling the definition of $M_\gamma^*(P_X)$ given in (2), define

$$\Delta = \sqrt{\frac{2M_\gamma^*(P_X) + n}{n(n-1)}} \cdot \sqrt{\mathbb{E}\left[Z\right]}.$$

We have

$$
\begin{aligned}
\sum_{m=1}^{M_\gamma^*(P_X)} p_m |\mu(x^{(m)}) - \mu_P(x^{(m)})| &= \sum_{m=1}^{M_\gamma^*(P_X)} \frac{p_m |\mu(x^{(m)}) - \mu_P(x^{(m)})|}{\sqrt{2 + np_m}} \cdot \sqrt{2 + np_m} \\
&\leq \sqrt{\sum_{m=1}^{M_\gamma^*(P_X)} \frac{p_m^2 (\mu(x^{(m)}) - \mu_P(x^{(m)}))^2}{2 + np_m}} \cdot \sqrt{\sum_{m=1}^{M_\gamma^*(P_X)} 2 + np_m} \\
&\leq \sqrt{\frac{\mathbb{E}\left[Z\right]}{n(n-1)}} \cdot \sqrt{2M_\gamma^*(P_X) + n} \\
&= \Delta,
\end{aligned}
$$

where the next-to-last step holds by the following identity:

**Lemma C.2.** *For all $n \geq 1$ and $p \in [0, 1]$, $np - 1 + (1-p)^n \geq \frac{n(n-1)p^2}{2+np}$.*

Next, we will use Lemma C.1 to relate $\Delta$ and $\widehat{\Delta}$. By Chebyshev's inequality, conditional on $X_1, \dots, X_n$, with probability at least $1 - \delta/4$ we have

$$Z \geq \mathbb{E}\left[Z \mid X_1, \dots, X_n\right] - \sqrt{\frac{\operatorname{Var}\left(Z \mid X_1, \dots, X_n\right)}{\delta/4}} \geq \mathbb{E}\left[Z \mid X_1, \dots, X_n\right] - \sqrt{\frac{N_{\geq 2} + 2\mathbb{E}\left[Z \mid X_1, \dots, X_n\right]}{\delta/4}},$$

which can be relaxed to

$$\mathbb{E}\left[Z \mid X_1, \dots, X_n\right] \leq 2Z + 4\sqrt{N_{\geq 2}/\delta} + 8/\delta.$$

Marginalizing over $X_1, \dots, X_n$, this bound holds with probability at least $1 - \delta/4$. Moreover, again applying Chebyshev's inequality, with probability at least $1 - \delta/4$ we have

$$\mathbb{E}\left[Z \mid X_1, \dots, X_n\right] \geq \mathbb{E}\left[Z\right] - \sqrt{\frac{\operatorname{Var}\left(\mathbb{E}\left[Z \mid X_1, \dots, X_n\right]\right)}{\delta/4}} \geq \mathbb{E}\left[Z\right] - \sqrt{\frac{2\mathbb{E}\left[Z\right]}{\delta/4}},$$

which can be relaxed to

$$\mathbb{E}\left[Z\right] \leq 2\mathbb{E}\left[Z \mid X_1, \dots, X_n\right] + 8/\delta.$$

Combining our bounds, then, we have $\mathbb{E}[Z] \leq 4Z + 8\sqrt{N_{\geq 2}/\delta} + 24/\delta$ with probability at least $1 - \delta/2$. Since $\mathbb{P}\left\{\widehat{M}_\gamma \geq M_\gamma^*(P_X)\right\} \geq 1 - \delta/2$ by Hoeffding's inequality, this implies that

$$\mathbb{P}\left\{\widehat{\Delta} \geq \Delta\right\} \geq 1 - \delta.$$

Now we verify the coverage properties of $\widehat{C}_n$. We have

$$\mathbb{P}\left\{\mu_P(X_{n+1}) \notin \widehat{C}_n(X_{n+1})\right\} = \mathbb{P}\left\{|\mu_P(X_{n+1}) - \mu(X_{n+1})| > (\alpha - \delta - \gamma)^{-1}\widehat{\Delta}\right\}$$

$$\leq \mathbb{P}\left\{\widehat{\Delta} < \Delta\right\} + \mathbb{P}\left\{|\mu_P(X_{n+1}) - \mu(X_{n+1})| > (\alpha - \delta - \gamma)^{-1}\Delta\right\}$$

$$\leq \mathbb{P}\left\{\widehat{\Delta} < \Delta\right\} + \mathbb{P}\left\{X_{n+1} \notin \{x^{(1)}, \ldots, x^{(M_\gamma^*(P_X))}\}\right\}$$

$$\qquad + \sum_{m=1}^{M_\gamma^*(P_X)} \mathbb{P}\left\{X_{n+1} = x^{(m)}, \, |\mu_P(X_{n+1}) - \mu(X_{n+1})| > (\alpha - \delta - \gamma)^{-1}\Delta\right\}$$

$$\leq \delta + \gamma + \sum_{m=1}^{M_\gamma^*(P_X)} \mathbb{P}\left\{X_{n+1} = x^{(m)}, \, |\mu_P(X_{n+1}) - \mu(X_{n+1})| > (\alpha - \delta - \gamma)^{-1}\Delta\right\}$$

$$\leq \delta + \gamma + \sum_{m=1}^{M_\gamma^*(P_X)} p_m \mathbb{1}\left\{\left|\mu_P(x^{(m)}) - \mu(x^{(m)})\right| > (\alpha - \delta - \gamma)^{-1}\Delta\right\}$$

$$\leq \delta + \gamma + \frac{\sum_{m=1}^{M_\gamma^*(P_X)} p_m \left|\mu_P(x^{(m)}) - \mu(x^{(m)})\right|}{(\alpha - \delta - \gamma)^{-1}\Delta}$$

$$\leq \delta + \gamma + \frac{\Delta}{(\alpha - \delta - \gamma)^{-1}\Delta} = \alpha,$$

which verifies the desired coverage guarantee.

## D   Proof of Theorem 3

First, we have $\widehat{M}_\gamma \leq M$ almost surely by our assumption on $P_X$. Next we need to bound $\mathbb{E}[Z_+]$. We have

$$\mathbb{E}[Z_-] \leq \mathbb{E}[(Z - \mathbb{E}[Z \mid X_1, \ldots, X_n])_-] \text{ since this conditional expectation is nonnegative}$$

$$\leq \sqrt{\mathbb{E}[(Z - \mathbb{E}[Z \mid X_1, \ldots, X_n])^2]}$$

$$= \sqrt{\mathbb{E}[\mathbb{E}[(Z - \mathbb{E}[Z \mid X_1, \ldots, X_n])^2 \mid X_1, \ldots, X_n]]}$$

$$= \sqrt{\mathbb{E}[\mathrm{Var}(Z \mid X_1, \ldots, X_n)]}$$

$$\leq \sqrt{\mathbb{E}[N_{\geq 2} + 2\mathbb{E}[Z \mid X_1, \ldots, X_n]]} \text{ by Lemma C.1}$$

$$= \sqrt{\mathbb{E}[N_{\geq 2}] + 2\mathbb{E}[Z]}.$$

We then have

$$\mathbb{E}[Z_+] = \mathbb{E}[Z] + \mathbb{E}[Z_-] \leq \mathbb{E}[Z] + \sqrt{2\mathbb{E}[Z] + \mathbb{E}[N_{\geq 2}]} \leq 1.5\mathbb{E}[Z] + 1 + \sqrt{\mathbb{E}[N_{\geq 2}]}.$$

Next we need a lemma:

**Lemma D.1.** *For all $n \geq 1$ and $p \in [0, 1]$, $np - 1 + (1 - p)^n \leq \frac{n^2 p^2}{1 + np}$.*

Combined with the calculation of $\mathbb{E}[Z]$ in Lemma C.1, we have

$$\mathbb{E}[Z] \leq \sum_{m=1}^{M} (\mu(x^{(m)}) - \mu_P(x^{(m)}))^2 \cdot \frac{n^2 p_m^2}{1 + n p_m}$$

$$\leq \sum_{m=1}^{M} p_m \cdot (\mu(x^{(m)}) - \mu_P(x^{(m)}))^2 \cdot \frac{n^2 \cdot \eta/M}{1 + n \cdot \eta/M}$$

$$= \frac{\eta n^2}{M + \eta n} \cdot \mathbb{E}_{P_X}\left[(\mu_P(X) - \mu(X))^2\right]$$

$$\leq (\mathrm{err}_\mu)^2 \cdot \frac{\eta n^2}{M + \eta n},$$

since we have assumed that $P_X$ is supported on $\{x^{(1)}, \ldots, x^{(M)}\}$ and that $\mathbb{P}_{P_X}\{X = x^{(m)}\} \leq \eta/M$ for all $m$, where we must have $\eta \geq 1$. Furthermore, we have

$$\mathbb{E}[N_{\geq 2}] = \sum_{m=1}^{M} \mathbb{P}\{n_m \geq 2\} \leq \sum_{m=1}^{M} \mathbb{E}[(n_m - 1)_+]$$

$$= \sum_{m=1}^{M} n \cdot \mathbb{P}_{P_X}\left\{X = x^{(m)}\right\} - 1 + \left(1 - \mathbb{P}_{P_X}\left\{X = x^{(m)}\right\}\right)^n \quad \text{as calculated as in the proof of Lemma C.1}$$

$$\leq \sum_{m=1}^{M} n \cdot \eta/M - 1 + (1 - \eta/M)^n$$

$$\leq \sum_{m=1}^{M} \frac{n^2 (\eta/M)^2}{1 + n\eta/M} \quad \text{by Lemma D.1}$$

$$= \frac{\eta^2 n^2}{M + \eta n}.$$

We also have $N_{\geq 2} \leq M$ almost surely, and so combining these two bounds, $\mathbb{E}[N_{\geq 2}] \leq \min\{\frac{\eta^2 n^2}{M}, M\}$. Combining everything, then,

$$\mathbb{E}[Z_+] \leq 1.5(\mathrm{err}_\mu)^2 \cdot \frac{\eta n^2}{M + \eta n} + 1 + \sqrt{\min\left\{\frac{\eta^2 n^2}{M}, M\right\}}.$$

Plugging these calculations into the definition of $\widehat{\Delta}$, we obtain

$$\mathbb{E}\left[\widehat{\Delta}\right] = \mathbb{E}\left[\sqrt{\frac{2\widehat{M}_\gamma + n}{n(n-1)}} \cdot \sqrt{4Z_+ + 8\sqrt{N_{\geq 2}/\delta} + 24/\delta}\right]$$

$$\leq \mathbb{E}\left[\sqrt{\frac{2M + n}{n(n-1)}} \cdot \sqrt{4Z_+ + 8\sqrt{N_{\geq 2}/\delta} + 24/\delta}\right]$$

$$\leq \sqrt{\frac{2M + n}{n(n-1)}} \cdot \sqrt{4\mathbb{E}[Z_+] + 8\sqrt{\mathbb{E}[N_{\geq 2}]/\delta} + 24/\delta}$$

$$\leq \sqrt{\frac{2M + n}{n(n-1)}} \cdot \sqrt{4\left(1.5(\mathrm{err}_\mu)^2 \cdot \frac{\eta n^2}{M + \eta n} + 1 + \sqrt{\min\left\{\frac{\eta^2 n^2}{M}, M\right\}}\right) + 8\sqrt{\min\left\{\frac{n^2}{M}, M\right\} \cdot 1/\delta} + 24/\delta}$$

$$\leq \sqrt{\frac{2M + n}{n(n-1)}} \cdot \left[\sqrt{6(\mathrm{err}_\mu)^2 \cdot \frac{\eta n^2}{M + \eta n}} + \sqrt{4(1 + 2/\sqrt{\delta})\sqrt{\min\left\{\frac{\eta^2 n^2}{M}, M\right\}} + \sqrt{4 + 24/\delta}}\right].$$

We can assume that $M \leq n^2$ and $n \geq 2$ (as otherwise, the upper bound would be trivial, since we must have $\mathrm{Leb}(\widehat{C}_n(X_{n+1})) \leq 1$ by construction). If $M \geq n$, then $\frac{2M+n}{n(n-1)} \leq \frac{6M}{n^2}$ and the above

simplifies to

$$\mathbb{E}\left[\widehat{\Delta}\right] \le 6\sqrt{\eta} \cdot \mathrm{err}_\mu + \sqrt{\frac{6(4+24/\delta)M}{n^2}} + \sqrt{24\eta(1+2/\sqrt{\delta})}\sqrt[4]{\frac{M}{n^2}},$$

and since we assume $M \le n^2$, we therefore have

$$\mathbb{E}\left[\widehat{\Delta}\right] \le 6\sqrt{\eta} \cdot \mathrm{err}_\mu + \left(\sqrt{6(4+24/\delta)} + \sqrt{24\eta(1+2/\sqrt{\delta})}\right) \cdot \sqrt[4]{\frac{M}{n^2}}. \tag{D.2}$$

If instead $M < n$, then $\frac{2M+n}{n(n-1)} \le \frac{6}{n}$ and the above bound on $\mathbb{E}\left[\widehat{\Delta}\right]$ simplifies to

$$\mathbb{E}\left[\widehat{\Delta}\right] \le 6 \cdot \mathrm{err}_\mu + \sqrt{\frac{6}{n}} \cdot \left[\sqrt{4(1+2/\sqrt{\delta})\sqrt{M}} + \sqrt{4+24/\delta}\right],$$

which again yields the same bound (D.2) since $M \ge 1$ and $\eta \ge 1$. Finally, by definition of $\widehat{C}_n(X_{n+1})$, we have

$$\mathbb{E}\left[\mathrm{Leb}(\widehat{C}_n(X_{n+1}))\right] \le \mathbb{E}\left[\widehat{\Delta}\right] \cdot \frac{2}{\alpha - \delta - \gamma},$$

which completes the proof for $c$ chosen appropriately as a function of $\alpha, \delta, \gamma, \eta$.

# E   Proofs of lemmas

## E.1   Proof of Lemma 1

Let $x_{\mathrm{med}}$ be the median of $Q$. Define

$$q_< = \mathbb{P}_Q\left\{X < x_{\mathrm{med}}\right\}, \ q_> = \mathbb{P}_Q\left\{X > x_{\mathrm{med}}\right\},$$

and note that $q_<, q_> \in [0, 0.5]$. For $X \sim Q$, let $Q_<$ be the distribution of $X$ conditional on $X < x_{\mathrm{med}}$ and let $Q_>$ be the distribution of $X$ conditional on $X > x_{\mathrm{med}}$. Then we can write

$$Q = q_< \cdot Q_< + (1 - q_< - q_>) \cdot \delta_{x_{\mathrm{med}}} + q_> \cdot Q_>,$$

where $\delta_t$ denotes the point mass distribution at $t$. Now define

$$Q_0 = 2q_< \cdot Q_< + (1 - 2q_<) \cdot \delta_{x_{\mathrm{med}}}$$

and

$$Q_1 = 2q_> \cdot Q_> + (1 - 2q_>) \cdot \delta_{x_{\mathrm{med}}}.$$

Then clearly $Q = 0.5Q_0 + 0.5Q_1$. Next let $\mu_0, \mu_1$ be the means of these two distributions, satisfying $\frac{\mu_0 + \mu_1}{2} = \mu$ where $\mu$ is the mean of $Q$, and let $\sigma_0^2, \sigma_1^2$ be the variances of these two distributions. By the law of total variance, we have

$$\sigma^2 = \mathrm{Var}\left(0.5\delta_{\mu_0} + 0.5\delta_{\mu_1}\right) + \mathbb{E}\left[0.5\delta_{\sigma_0^2} + 0.5\delta_{\sigma_1^2}\right]$$

$$= \frac{(\mu_1 - \mu_0)^2}{4} + 0.5\sigma_0^2 + 0.5\sigma_1^2.$$

Next, $Q_0$ is a distribution supported on $[0, x_{\mathrm{med}}]$ with mean $\mu_0$, so its variance is bounded as

$$\sigma_0^2 \le \mu_0(x_{\mathrm{med}} - \mu_0),$$

where the maximum is attained if all the mass is placed on the endpoints $0$ or $x_{\mathrm{med}}$. Similarly, $Q_1$ is a distribution supported on $[x_{\mathrm{med}}, 1]$ with mean $\mu_1$, so its variance is bounded as

$$\sigma_1^2 \le (1 - \mu_1)(\mu_1 - x_{\mathrm{med}}).$$

Using the fact that $\frac{\mu_0 + \mu_1}{2} = \mu$, we can simplify to

$$\sigma_0^2 + \sigma_1^2 \le \mu_0(x_{\mathrm{med}} - \mu_0) + (1 - \mu_1)(\mu_1 - x_{\mathrm{med}})$$

$$= \mu(x_{\mathrm{med}} - \mu_0) + (1 - \mu)(\mu_1 - x_{\mathrm{med}}) - 0.5(\mu_1 - \mu_0)^2.$$

Therefore, we have

$$\sigma^2 = \frac{(\mu_1 - \mu_0)^2}{4} + 0.5\sigma_0^2 + 0.5\sigma_1^2 \le 0.5\mu(x_{\mathrm{med}} - \mu_0) + 0.5(1 - \mu)(\mu_1 - x_{\mathrm{med}})$$

$$= 0.5(2\mu - 1)x_{\mathrm{med}} - 0.5\mu\mu_0 + 0.5(1 - \mu)\mu_1 = 0.5(2\mu - 1)(x_{\mathrm{med}} - \mu) + 0.25(\mu_1 - \mu_0).$$

Next, $|2\mu - 1| \le 1$ since $\mu \in [0, 1]$, and $|x_{\mathrm{med}} - \mu| \le 0.5|\mu_1 - \mu_0|$ since $\mu_0 \le x_{\mathrm{med}} \le \mu_1$ and $\frac{\mu_0 + \mu_1}{2} = \mu$. Therefore, $\sigma^2 \le 0.5(\mu_1 - \mu_0)$, proving the lemma.

## E.2 Proof of Lemma 2

First we need a supporting lemma.

**Lemma E.1.** *For any $N \geq 1$ and any $\epsilon \in [0, 0.5]$,*

$$d_{\mathrm{KL}}\Big(0.5 \cdot \mathrm{Binom}(N, 0.5 + \epsilon) + 0.5 \cdot \mathrm{Binom}(N, 0.5 - \epsilon) \,\big\|\, \mathrm{Binom}(N, 0.5)\Big) \leq 8N(N-1)\epsilon^4.$$

*Proof of Lemma E.1.* Let $f_0$ be the probability mass function of the $\mathrm{Binom}(N, 0.5)$ distribution, and let $f_1$ be the probability mass function of the mixture $0.5 \cdot \mathrm{Binom}(N, 0.5 + \epsilon) + 0.5 \cdot \mathrm{Binom}(N, 0.5 - \epsilon)$. Then we would like to bound $d_{\mathrm{KL}}(f_1 \| f_0)$. We calculate the ratio

$$\frac{f_1(k)}{f_0(k)} = \frac{0.5 \cdot \binom{N}{k}(0.5 + \epsilon)^k(0.5 - \epsilon)^{N-k} + 0.5 \cdot \binom{N}{k}(0.5 - \epsilon)^k(0.5 + \epsilon)^{N-k}}{\binom{N}{k}(0.5)^N}$$

$$= \frac{(1 + 2\epsilon)^k(1 - 2\epsilon)^{N-k} + (1 - 2\epsilon)^k(1 + 2\epsilon)^{N-k}}{2}.$$

Therefore, it holds that

$$\mathbb{E}_{\mathrm{Binom}(N,0.5)}\left[\left(\frac{f_1(X)}{f_0(X)}\right)^2\right]$$

$$= \mathbb{E}_{\mathrm{Binom}(N,0.5)}\left[\left(\frac{(1 + 2\epsilon)^X(1 - 2\epsilon)^{N-X} + (1 - 2\epsilon)^X(1 + 2\epsilon)^{N-X}}{2}\right)^2\right]$$

$$= \mathbb{E}_{\mathrm{Binom}(N,0.5)}\left[\frac{(1 + 2\epsilon)^{2X}(1 - 2\epsilon)^{2N-2X} + (1 - 2\epsilon)^{2X}(1 + 2\epsilon)^{2N-2X} + 2(1 - 4\epsilon^2)^N}{4}\right]$$

$$= \frac{(1 - 2\epsilon)^{2N}\mathbb{E}_{\mathrm{Binom}(N,0.5)}\left[\left(\frac{1+2\epsilon}{1-2\epsilon}\right)^{2X}\right] + (1 + 2\epsilon)^{2N}\mathbb{E}_{\mathrm{Binom}(N,0.5)}\left[\left(\frac{1-2\epsilon}{1+2\epsilon}\right)^{2X}\right] + 2(1 - 4\epsilon^2)^N}{4}$$

$$= \frac{(1 - 2\epsilon)^{2N}\mathbb{E}_{\mathrm{Bern}(0.5)}\left[\left(\frac{1+2\epsilon}{1-2\epsilon}\right)^{2X}\right]^N + (1 + 2\epsilon)^{2N}\mathbb{E}_{\mathrm{Bern}(0.5)}\left[\left(\frac{1-2\epsilon}{1+2\epsilon}\right)^{2X}\right]^N + 2(1 - 4\epsilon^2)^N}{4}$$

$$= \frac{(1 - 2\epsilon)^{2N}\left[0.5\left(\frac{1+2\epsilon}{1-2\epsilon}\right)^2 + 0.5\right]^N + (1 + 2\epsilon)^{2N}\left[0.5\left(\frac{1-2\epsilon}{1+2\epsilon}\right)^2 + 0.5\right]^N + 2(1 - 4\epsilon^2)^N}{4}$$

$$= \frac{\left[0.5(1 + 2\epsilon)^2 + 0.5(1 - 2\epsilon)^2\right]^N + \left[0.5(1 - 2\epsilon)^2 + 0.5(1 + 2\epsilon)^2\right]^N + 2(1 - 4\epsilon^2)^N}{4}$$

$$= \frac{(1 + 4\epsilon^2)^N + (1 - 4\epsilon^2)^N}{2}$$

$$= 1 + \sum_{k \geq 1}\binom{N}{2k}(4\epsilon^2)^{2k}$$

$$= 1 + \sum_{k \geq 1}\frac{N(N-1)\ldots(N - 2k + 2)(N - 2k + 1)}{(2k)!}(4\epsilon^2)^{2k}$$

$$\leq 1 + \sum_{k \geq 1}\frac{(N(N-1))^k}{2^k k!}(4\epsilon^2)^{2k}$$

$$\leq e^{8\epsilon^4 N(N-1)}.$$

Applying Jensen's inequality, we then have

$$d_{\mathrm{KL}}(f_1 \| f_0) = \sum_{k=0}^{n} f_1(k)\log\left(\frac{f_1(k)}{f_0(k)}\right) = \mathbb{E}_{f_1}\left[\log\left(\frac{f_1(X)}{f_0(X)}\right)\right] \leq \log\left(\mathbb{E}_{f_1}\left[\frac{f_1(X)}{f_0(X)}\right]\right)$$

$$= \log\left(\mathbb{E}_{\mathrm{Binom}(N,0.5)}\left[\left(\frac{f_1(X)}{f_0(X)}\right)^2\right]\right) \leq \log\left(e^{8\epsilon^4 N(N-1)}\right) = 8\epsilon^4 N(N-1).$$

$\square$

Now we turn to the proof of Lemma 2. Let $p_m = \mathbb{P}\{X \in \mathcal{X}_m\}$ for each $m = 1, 2, \ldots$. Define a distribution $P_0'$ on $(W, Z) \in \mathbb{N} \times \{0, 1\}$ as:

$$\text{Draw } W \sim \sum_{m=1}^{\infty} p_m \delta_m, \text{ and draw } Z \sim \text{Bernoulli}(0.5), \text{ independently from } W.$$

and for any signs $a_1, a_2, \cdots \in \{\pm 1\}$, define a distribution $P_a'$ on $(W, Z) \in \mathbb{N} \times \{0, 1\}$ as:

$$\text{Draw } W \sim \sum_{m=1}^{\infty} p_m \delta_m, \text{ and conditional on } W, \text{ draw } Z|W = m \sim \text{Bernoulli}(0.5 + a_m \cdot \epsilon_m).$$

Then define $\tilde{P}_0' = (P_0')^n$ and define $\tilde{P}_1'$ as the following mixture distribution.

- Draw $A_1, A_2, \ldots \overset{\text{iid}}{\sim} \text{Unif}\{\pm 1\}$.

- Conditional on $A_1, A_2, \ldots$, draw $(W_1, Z_1), \ldots, (W_n, Z_n) \overset{\text{iid}}{\sim} P_A'$.

Note that $(X_1, Z_1), \ldots, (X_n, Z_n) \sim \tilde{P}_0$ can be drawn by first drawing $(W_1, Z_1), \ldots, (W_n, Z_n) \sim \tilde{P}_0'$ and then drawing $X_i|W_i \sim P_{X|X \in \mathcal{X}_{W_i}}$ for each $i$. Similarly, $(X_1, Z_1), \ldots, (X_n, Z_n) \sim \tilde{P}_1$ is equivalent to first drawing $(W_1, Z_1), \ldots, (W_n, Z_n) \sim \tilde{P}_1'$ and then drawing $X_i|W_i \sim P_{X|X \in \mathcal{X}_{W_i}}$ for each $i$. This implies $d_{\text{TV}}(\tilde{P}_1||\tilde{P}_0) \leq d_{\text{TV}}(\tilde{P}_1'||\tilde{P}_0')$.

Now we can calculate the probability mass function of $\tilde{P}_0'$ as

$$\tilde{P}_0'\big((w_1, z_1), \ldots, (w_n, z_n)\big) = \prod_{i=1}^{n} (p_{w_i} \cdot 0.5),$$

and for $\tilde{P}_1'$ as

$$\tilde{P}_1'\big((w_1, z_1), \ldots, (w_n, z_n)\big) = \mathbb{E}_{A_i \overset{\text{iid}}{\sim} \text{Unif}\{\pm 1\}} \left[ \prod_{i=1}^{n} \left( p_{w_i} \cdot (0.5 + A_{w_i} \epsilon_m)^{z_i} \cdot (0.5 - A_{w_i} \epsilon_m)^{1-z_i} \right) \right].$$

Defining summary statistics

$$n_m = \sum_{i=1}^{n} \mathbb{1}\{w_i = m\} \text{ and } k_m = \sum_{i=1}^{n} \mathbb{1}\{w_i = m, z_i = 1\},$$

we can rewrite the above as

$$\tilde{P}_0'\big((w_1, z_1), \ldots, (w_n, z_n)\big) = \prod_{m=1}^{\infty} p_m^{n_m} \cdot 0.5^{n_m},$$

and

$$\tilde{P}_1'\big((w_1, z_1), \ldots, (w_n, z_n)\big) = \mathbb{E}_{A_i \overset{\text{iid}}{\sim} \text{Unif}\{\pm 1\}} \left[ \prod_{m=1}^{\infty} p_m^{n_m} \cdot (0.5 + A_m \epsilon_m)^{k_m} \cdot (0.5 - A_m \epsilon_m)^{n_m - k_m} \right]$$

$$= \prod_{m=1}^{\infty} p_m^{n_m} \cdot \frac{1}{2} \sum_{a_m \in \{\pm 1\}} (0.5 + a_m \epsilon_m)^{k_m} \cdot (0.5 - a_m \epsilon_m)^{n_m - k_m}$$

We then calculate

$$
\begin{aligned}
d_{\mathrm{KL}}(\tilde{P}_1'||\tilde{P}_0') &= \mathbb{E}_{\tilde{P}_1}\left[\log\left(\frac{\tilde{P}_1'\big((W_1, Z_1), \ldots, (W_n, Z_n)\big)}{\tilde{P}_0'\big((W_1, Z_1), \ldots, (W_n, Z_n)\big)}\right)\right] \\
&= \mathbb{E}_{\tilde{P}_1'}\left[\log\left(\frac{\prod_{m=1}^{\infty} p_m^{N_m} \cdot \frac{1}{2}\sum_{a_m \in \{\pm 1\}}(0.5 + a_m\epsilon_m)^{K_m} \cdot (0.5 - a_m\epsilon_m)^{N_m - K_m}}{\prod_{m=1}^{\infty} p_m^{N_m} \cdot (0.5)^{N_m}}\right)\right] \\
&= \sum_{m=1}^{\infty} \mathbb{E}_{\tilde{P}_1'}\left[\log\left(\frac{\frac{1}{2}\sum_{a_m \in \{\pm 1\}}(0.5 + a_m\epsilon_m)^{K_m} \cdot (0.5 - a_m\epsilon_m)^{N_m - K_m}}{(0.5)^{N_m}}\right)\right] \\
&= \sum_{m=1}^{\infty} \mathbb{E}_{\tilde{P}_1'}\left[\mathbb{E}_{\tilde{P}_1'}\left[\log\left(\frac{\frac{1}{2}\sum_{a_m \in \{\pm 1\}}(0.5 + a_m\epsilon_m)^{K_m} \cdot (0.5 - a_m\epsilon_m)^{N_m - K_m}}{(0.5)^{N_m}}\right)\,\bigg|\, N_m\right]\right],
\end{aligned}
$$

where

$$
N_m = \sum_{i=1}^{n} \mathbb{1}\{W_i = m\} \text{ and } K_m = \sum_{i=1}^{n} \mathbb{1}\{W_i = m, Z_i = 1\},
$$

Next, we calculate the conditional expectation in the last expression above. If $N_m = 0$ then trivially it is equal to $\log(1) = 0$. If $N_m \geq 1$, then under $\tilde{P}_1'$, we can see that

$$
K_m \mid N_m \sim 0.5 \cdot \mathrm{Binom}(N_m, 0.5 + \epsilon_m) + 0.5 \cdot \mathrm{Binom}(N_m, 0.5 - \epsilon_m),
$$

and therefore,

$$
\begin{aligned}
&\mathbb{E}_{\tilde{P}_1'}\left[\log\left(\frac{\frac{1}{2}\sum_{a_m \in \{\pm 1\}}(0.5 + a_m\epsilon_m)^{K_m} \cdot (0.5 - a_m\epsilon_m)^{N_m - K_m}}{(0.5)^{N_m}}\right)\,\bigg|\, N_m\right] \\
&= d_{\mathrm{KL}}\Big(0.5 \cdot \mathrm{Binom}(N_m, 0.5 + \epsilon_m) + 0.5 \cdot \mathrm{Binom}(N_m, 0.5 - \epsilon_m) \,\big\|\, \mathrm{Binom}(N_m, 0.5)\Big) \leq 8 N_m(N_m - 1)\epsilon_m^4,
\end{aligned}
$$

where the last step applies Lemma E.1. Therefore,

$$
\begin{aligned}
d_{\mathrm{KL}}(\tilde{P}_1'||\tilde{P}_0') &\leq \sum_{m=1}^{\infty} \mathbb{E}_{\tilde{P}_1'}\left[8 N_m(N_m - 1)\epsilon_m^4\right] \\
&= 8\sum_{m=1}^{\infty} \epsilon_m^4 \mathbb{E}_{\tilde{P}_1'}\left[N_m^2 - N_m\right] \\
&= 8\sum_{m=1}^{\infty} \epsilon_m^4 \left(\big(np_m(1 - p_m) + n^2 p_m^2\big) - np_m\right) \\
&= 8 \cdot n(n - 1)\sum_{m=1}^{\infty} \epsilon_m^4 p_m^2,
\end{aligned}
$$

since $N_m \sim \mathrm{Binom}(n, p_m)$ by definition. Applying Pinsker's inequality and $d_{\mathrm{TV}}(\tilde{P}_1||\tilde{P}_0) \leq d_{\mathrm{TV}}(\tilde{P}_1'||\tilde{P}_0')$ completes the proof.

### E.3 Proof of Lemma C.1

Define

$$
Z_m = \begin{cases} (n_m - 1) \cdot \big((\bar{y}_m - \mu(x^{(m)}))^2 - n_m^{-1} s_m^2\big), & n_m \geq 2, \\ 0, & n_m = 0 \text{ or } 1. \end{cases}
$$

Then $Z = \sum_{m=1}^{\infty} Z_m$. Now we calculate the conditional mean and variance. Conditional on $X_1, \ldots, X_n$, $\bar{y}_m$ and $s_m^2$ are the sample mean and sample variance of $n_m$ i.i.d. draws from a distribution with mean $\mu_P(x^{(m)})$ and variance $\sigma_P^2(x^{(m)})$, supported on $[0, 1]$, where we let $\sigma_P^2(x^{(m)})$ be the variance of the distribution of $Y|X = x^{(m)}$, under the joint distribution $P$. For any $m$ with $n_m \geq 2$, we therefore have

$$
\mathbb{E}\left[\bar{y}_m \mid X_1, \ldots, X_n\right] = \mu_P(x^{(m)}), \; \mathrm{Var}\left(\bar{y}_m \mid X_1, \ldots, X_n\right) = n_m^{-1}\sigma_P^2(x^{(m)}) = \mathbb{E}\left[n_m^{-1} s_m^2 \mid X_1, \ldots, X_n\right],
$$

and so

$$\mathbb{E}\left[(\bar{y}_m - \mu(x^{(m)}))^2 - n_m^{-1}s_m^2 \,\Big|\, X_1, \ldots, X_n\right]$$
$$= n_m^{-1}\sigma_P^2(x^{(m)}) + (\mu_P(x^{(m)}) - \mu(x^{(m)}))^2 - n_m^{-1}\sigma_P^2(x^{(m)}) = (\mu_P(x^{(m)}) - \mu(x^{(m)}))^2.$$

Next, we have $(n_1, \ldots, n_M) \sim \text{Multinom}(n, p)$, which implies that marginally $n_m \sim \text{Binom}(n, p_m)$ and so

$$\mathbb{E}\left[(n_m - 1)_+\right] = \mathbb{E}\left[n_m - 1 + \mathbb{1}\{n_m = 0\}\right] = np_m - 1 + (1 - p_m)^n.$$

Combining these calculations completes the proof for the expected value $\mathbb{E}[Z]$ and conditional expected value $\mathbb{E}[Z \mid X_1, \ldots, X_n]$.

Next, we calculate conditional and marginal variance. We have

$$\text{Var}\left((\bar{y}_m - \mu(x^{(m)}))^2 - n_m^{-1}s_m^2 \,\Big|\, X_1, \ldots, X_n\right)$$
$$= \text{Var}\left((\bar{y}_m - \mu(x^{(m)}))^2 - n_m^{-1}s_m^2 - (\mu_P(x^{(m)}) - \mu(x^{(m)}))^2 \,\Big|\, X_1, \ldots, X_n\right)$$
$$\leq \mathbb{E}\left[\left((\bar{y}_m - \mu(x^{(m)}))^2 - n_m^{-1}s_m^2 - (\mu_P(x^{(m)}) - \mu(x^{(m)}))^2\right)^2 \,\Big|\, X_1, \ldots, X_n\right]$$
$$= \mathbb{E}\left[\left((\bar{y}_m - \mu_P(x^{(m)}))^2 + 2(\bar{y}_m - \mu_P(x^{(m)}))(\mu_P(x^{(m)}) - \mu(x^{(m)})) - n_m^{-1}s_m^2\right)^2 \,\Big|\, X_1, \ldots, X_n\right]$$
$$\leq 4\mathbb{E}\left[\left((\bar{y}_m - \mu_P(x^{(m)}))\right)^4 \,\Big|\, X_1, \ldots, X_n\right]$$
$$+ 2\mathbb{E}\left[\left(2(\bar{y}_m - \mu_P(x^{(m)}))(\mu_P(x^{(m)}) - \mu(x^{(m)}))\right)^2 \,\Big|\, X_1, \ldots, X_n\right]$$
$$+ 4\mathbb{E}\left[\left(n_m^{-1}s_m^2\right)^2 \,\Big|\, X_1, \ldots, X_n\right],$$

where the last step holds since $(a + b + c)^2 \leq 4a^2 + 2b^2 + 4c^2$ for any $a, b, c$. Now we bound each term separately. First, we have

$$\mathbb{E}\left[\left((\bar{y}_m - \mu_P(x^{(m)}))\right)^4 \,\Big|\, X_1, \ldots, X_n\right]$$
$$= \frac{1}{n_m^4} \sum_{\substack{i_1, i_2, i_3, i_4 \text{ s.t.} \\ X_{i_1} = X_{i_2} = X_{i_3} = X_{i_4} = x^{(m)}}} \mathbb{E}\left[\prod_{k=1}^4 (Y_{i_k} - \mu_P(x^{(m)})) \,\Big|\, X_1, \ldots, X_n\right]$$
$$= \frac{1}{n_m^4}\left[n_m \cdot \mathbb{E}\left[(Y - \mu_P(x^{(m)}))^4 \,\Big|\, X = x^{(m)}\right] + 3n_m(n_m - 1) \cdot \mathbb{E}\left[(Y - \mu_P(x^{(m)}))^2 \,\Big|\, X = x^{(m)}\right]^2\right]$$
$$\leq \frac{1}{n_m^4}\left[n_m \cdot \sigma_P^2(x^{(m)}) + 3n_m(n_m - 1) \cdot (\sigma_P^2(x^{(m)}))^2\right]$$
$$\leq \frac{1}{n_m^4}\left[n_m \cdot \tfrac{1}{4} + 3n_m(n_m - 1) \cdot (\tfrac{1}{4})^2\right] = \frac{3n_m + 1}{16n_m^3},$$

where the second step holds by counting tuples $(i_1, i_2, i_3, i_4)$ where either all four indices are equal, or there are two pairs of equal indices (since otherwise, the expected value of the product is zero). Next,

$$\mathbb{E}\left[\left(2(\bar{y}_m - \mu_P(x^{(m)}))(\mu_P(x^{(m)}) - \mu(x^{(m)}))\right)^2 \,\Big|\, X_1, \ldots, X_n\right]$$
$$= 4(\mu_P(x^{(m)}) - \mu(x^{(m)}))^2 \mathbb{E}\left[(\bar{y}_m - \mu_P(x^{(m)}))^2 \,\Big|\, X_1, \ldots, X_n\right]$$
$$= 4(\mu_P(x^{(m)}) - \mu(x^{(m)}))^2 \cdot n_m^{-1}\sigma_P^2(x^{(m)})$$
$$\leq n_m^{-1}(\mu_P(x^{(m)}) - \mu(x^{(m)}))^2.$$

Finally, since $s_m^2 \leq \frac{n_m}{4(n_m-1)}$ holds deterministically,

$$\mathbb{E}\left[\left(n_m^{-1}s_m^2\right)^2 \mid X_1,\ldots,X_n\right] \leq n_m^{-2} \cdot \frac{n_m}{4(n_m-1)} \cdot \mathbb{E}\left[s_m^2 \mid X_1,\ldots,X_n\right]$$

$$= n_m^{-2} \cdot \frac{n_m}{4(n_m-1)} \cdot \sigma_P^2(x^{(m)}) \leq \frac{1}{16n_m(n_m-1)}.$$

Combining everything, then,

$$\mathrm{Var}\left((\bar{y}_m - \mu(x^{(m)}))^2 - n_m^{-1}s_m^2 \mid X_1,\ldots,X_n\right)$$
$$\leq 4 \cdot \frac{3n_m+1}{16n_m^3} + 2 \cdot n_m^{-1}(\mu_P(x^{(m)}) - \mu(x^{(m)}))^2 + 4 \cdot \frac{1}{16n_m(n_m-1)},$$

and so for $n_m \geq 2$,

$$\mathrm{Var}\left(Z_m \mid X_1,\ldots,X_n\right)$$
$$\leq (n_m-1)^2 \cdot \left[4 \cdot \frac{3n_m+1}{16n_m^3} + 2 \cdot n_m^{-1}(\mu_P(x^{(m)}) - \mu(x^{(m)}))^2 + 4 \cdot \frac{1}{16n_m(n_m-1)}\right]$$
$$\leq 1 + 2(n_m-1) \cdot (\mu_P(x^{(m)}) - \mu(x^{(m)}))^2 = 0.5 + 2\mathbb{E}\left[Z_m \mid X_1,\ldots,X_n\right].$$

If instead $n_m = 0$ or $n_m = 1$ then $Z_m = 0$ by definition, and so $\mathrm{Var}\left(Z_m \mid X_1,\ldots,X_n\right) = 0$. Therefore, in all cases, we have

$$\mathrm{Var}\left(Z_m \mid X_1,\ldots,X_n\right) \leq \mathbb{1}\left\{n_m \geq 2\right\} + 2\mathbb{E}\left[Z_m \mid X_1,\ldots,X_n\right].$$

It is also clear that, conditional on $X_1,\ldots,X_n$, the $Z_m$'s are independent, and so

$$\mathrm{Var}\left(Z \mid X_1,\ldots,X_n\right) = \sum_{m=1}^{\infty} \mathrm{Var}\left(Z_m \mid X_1,\ldots,X_n\right) \leq N_{\geq 2} + 2\mathbb{E}\left[Z \mid X_1,\ldots,X_n\right].$$

Finally, we need to bound $\mathrm{Var}\left(\mathbb{E}\left[Z \mid X_1,\ldots,X_n\right]\right)$. First, we have

$$\mathrm{Var}\left(\mathbb{E}\left[Z_m \mid X_1,\ldots,X_n\right]\right) = \mathrm{Var}\left((n_m-1)_+\right) \cdot (\mu_P(x^{(m)}) - \mu(x^{(m)}))^4$$
$$\leq \mathrm{Var}\left((n_m-1)_+\right) \cdot (\mu_P(x^{(m)}) - \mu(x^{(m)}))^2,$$

and we can calculate

$\mathrm{Var}\left((n_m-1)_+\right)$
$= \mathrm{Var}\left(n_m + \mathbb{1}\left\{n_m = 0\right\}\right)$
$= \mathrm{Var}\left(n_m\right) + \mathrm{Var}\left(\mathbb{1}\left\{n_m = 0\right\}\right) + 2\mathrm{Cov}\left(n_m, \mathbb{1}\left\{n_m = 0\right\}\right)$
$= \mathrm{Var}\left(n_m\right) + \mathrm{Var}\left(\mathbb{1}\left\{n_m = 0\right\}\right) - 2\mathbb{E}\left[n_m\right]\mathbb{E}\left[\mathbb{1}\left\{n_m = 0\right\}\right]$ since $n_m \cdot \mathbb{1}\left\{n_m = 0\right\} = 0$ almost surely
$= np_m(1-p_m) + (1-p_m)^n\left(1 - (1-p_m)^n\right) - 2np_m(1-p_m)^n.$

Therefore,

$2\mathbb{E}\left[(n_m-1)_+\right] - \mathrm{Var}\left((n_m-1)_+\right)$
$= 2np_m - 2 + 2(1-p_m)^n - np_m(1-p_m) - (1-p_m)^n\left(1 - (1-p_m)^n\right) + 2np_m(1-p_m)^n$
$= np_m(1+p_m) + (1-p_m)^n\left(1 + 2np_m + (1-p_m)^n\right) - 2$
$\geq 0,$

where the last step holds since, defining $f(t) = nt(1+t) + (1-t)^n\left(1 + 2nt + (1-t)^n\right)$, we can see that $f(0) = 2$ and $f'(t) \geq 0$ for all $t \in [0,1]$. This verifies that

$$\mathrm{Var}\left(\mathbb{E}\left[Z_m \mid X_1,\ldots,X_n\right]\right) \leq \mathrm{Var}\left((n_m-1)_+\right) \cdot (\mu_P(x^{(m)}) - \mu(x^{(m)}))^2$$
$$\leq 2\mathbb{E}\left[(n_m-1)_+\right] \cdot (\mu_P(x^{(m)}) - \mu(x^{(m)}))^2 = 2\mathbb{E}\left[Z_m\right].$$

Next, for any $m \neq m'$,

$$\text{Cov}\left(\mathbb{E}\left[Z_m \mid X_1, \ldots, X_n\right], \mathbb{E}\left[Z_{m'} \mid X_1, \ldots, X_n\right]\right)$$

$$= \text{Cov}\left((n_m - 1)_+, (n_{m'} - 1)_+\right) \cdot (\mu_P(x^{(m)}) - \mu(x^{(m)}))^2 \cdot (\mu_P(x^{(m')}) - \mu(x^{(m')}))^2$$

$$\leq 0.$$

For the last step, we use the fact that $\text{Cov}\left((n_m - 1)_+, (n_{m'} - 1)_+\right) \leq 0$, which holds since, conditional on $n_m$, we have $n_{m'} \sim \text{Binom}\left(n - n_m, \frac{p_{m'}}{1 - p_m}\right)$, and so the distribution of $n_{m'}$ is stochastically smaller whenever $n_m$ is larger. Therefore,

$$\text{Var}\left(\mathbb{E}\left[Z \mid X_1, \ldots, X_n\right]\right) \leq \sum_{m=1}^{\infty} \text{Var}\left(\mathbb{E}\left[Z_m \mid X_1, \ldots, X_n\right]\right) \leq \sum_{m=1}^{\infty} 2\mathbb{E}\left[Z_m\right] = 2\mathbb{E}\left[Z\right].$$

## E.4 Proofs of Lemma C.2 and Lemma D.1

Replacing $p$ with $1 - s$, equivalently, we need to show that, for all $s \in [0, 1]$,

$$\frac{n(n-1)(1-s)^2}{2 + n(1-s)} \leq n(1-s) - 1 + s^n \leq \frac{n^2(1-s)^2}{1 + n(1-s)}.$$

After simplifying, this is equivalent to proving that

$$\frac{n(1-s)^2 + 2n(1-s)}{2 + n(1-s)} \geq 1 - s^n \geq \frac{n(1-s)}{1 + n(1-s)},$$

which we can further simplify to

$$\frac{n(1-s) + 2n}{2 + n(1-s)} \geq 1 + s + \cdots + s^{n-1} \geq \frac{n}{1 + n(1-s)} \qquad (\text{E.2})$$

by dividing by $1 - s$ (note that this division can be performed whenever $s < 1$, while if $s = 1$, then the desired inequalities hold trivially).

Now we address the two desired inequalities separately. For the left-hand inequality in (E.2), define

$$h(s) = (2 + n(1-s)) \cdot (s + s^2 + \cdots + s^{n-1}) = ns + 2(s + s^2 + \cdots + s^{n-1}) - ns^n.$$

We calculate $h(1) = 2(n-1)$, and for any $s \in [0, 1]$,

$$h'(s) = n + \sum_{i=1}^{n-1} 2is^{i-1} - n^2 s^{n-1} \geq n + \sum_{i=1}^{n-1} 2is^{n-1} - n^2 s^{n-1}$$

$$= n + s^{n-1}\left(\sum_{i=1}^{n-1} 2i - n^2\right) = n - ns^{n-1} \geq 0,$$

where the first inequality holds since $s^{i-1} \geq s^{n-1}$ for all $i = 1, \ldots, n-1$, and the second inequality holds since $s^{n-1} \leq 1$. Therefore, $h(s) \leq h(1) = 2(n-1)$ for all $s \in [0, 1]$, and so

$$1 + s + \cdots + s^{n-1} = \frac{(1 + s + \cdots + s^{n-1}) \cdot (2 + n(1-s))}{2 + n(1-s)}$$

$$= \frac{2 + n(1-s) + h(s)}{2 + n(1-s)} \leq \frac{2 + n(1-s) + 2(n-1)}{2 + n(1-s)} = \frac{n(1-s) + 2n}{2 + n(1-s)},$$

as desired.

To verify the right-hand inequality in (E.2), we have

$$1 + s + \cdots + s^{n-1} = \frac{(1 + s + \cdots + s^{n-1}) \cdot (1 + n(1-s))}{1 + n(1-s)}$$

$$= \frac{(n+1)(1 + s + \cdots + s^{n-1}) - n(s + s^2 + \cdots + s^n)}{1 + n(1-s)}$$

$$= \frac{n + (1 + s + \cdots + s^{n-1}) - ns^n}{1 + n(1-s)}$$

$$\geq \frac{n}{1 + n(1-s)},$$

where the last step holds since, for $s \in [0, 1]$, we have $s^i \geq s^n$ for all $i = 0, 1, \ldots, n-1$.

# References

Siu-On Chan, Ilias Diakonikolas, Paul Valiant, and Gregory Valiant. Optimal algorithms for testing closeness of discrete distributions. In *Proceedings of the twenty-fifth annual ACM-SIAM symposium on Discrete algorithms*, pages 1193–1203. SIAM, 2014.