# OpenReview forum: "Distribution-free inference for regression: discrete, continuous, and in between"
_NeurIPS.cc/2021/Conference — NeurIPS 2021 Poster_

### Official Review · Reviewer_tn9d · 2021-07-12

**Rating:** 8
**Confidence:** 4

**Summary:**

This paper studies the construction of model-free confidence intervals for the conditional mean (regression function) of some outcome given a set of observed explanatory variables; i.e., the problem of estimating nonparametric regression functions. This problem is known to be very hard in general---it admits no non-trivial solution---but this paper makes interesting advances that are relevant for the special cases in which the distribution of the explanatory variables is mostly comprised of point masses (think of a discrete distribution, for example). In particular, a new lower bound is derived for the length of any model-free confidence interval with marginal coverage for the regression function, and a concrete algorithm based on data splitting that can achieve this bound is developed.

**Limitations And Societal Impact:**

The main limitation of this work is its lack of practical demonstrations, applications, and its dubious practical relevance. These are not necessarily major flaws, as the theoretical content is sufficiently novel and non-trivial to justify acceptance, in my opinion. However, my impression is that this theoretical effort is ultimately motivated by its potential practical implications for machine learning and statistics practitioners, but such practical aspects are completely omitted.

**Main Review:**

This work studies an interesting and novel variation of a well-known problem with broad relevance, and it develops an original solution for it. Overall, the paper is very well written and contains elegant mathematical results. These theoretical results intuitively make sense and their proofs appear to be technically correct.
It must be noted that this is a purely theoretical paper; it contains no numerical experiments, no comparisons to alternative approaches, and no applications to real data. In fact not much effort at all is spent in justifying the practical relevance of this work, which is not completely self evident to me.

Major comments.
 - A possible limitation is the dubious practical relevance of this work because the proposed algorithm will only really be useful in cases where the explanatory variables can only take a small number of possible values compared to the sample size. In particular, this rules out all high-dimensional applications and all low-dimensional applications with continuous-valued variables. What concrete applications of interest to the machine learning community are left then? See also the following section on "Limitations And Societal Impact".
- What happens if the distribution of X is discrete but the dimensions d increase? I expect the upper bound should become trivial, as this problem seems very hard in high dimensions, and the algorithm relies on observing several identical values of X, which we know is extremely unlikely to occur in high dimensions. Perhaps a simple analytical example of this would be informative at the end of Section 3.

Minor comments:
 - I find the title to be not very informative, and perhaps even a little confusing. In my opinion is does not convey clearly the idea the this work is about the construction of model-free confidence intervals for the conditional mean (model-free inference is typically predictive) and it does not suggest that it focuses on discrete explanatory variables (instead, I first thought the paper would be about classification).
 - If this work has any concrete practical relevance it should be emphasized, perhaps by providing some examples of how it could be useful, and some comparisons of its performance against that of existing alternatives (possibly parametric ones, if non-parametric alternatives do not exist). If no meaningful applications come to mind, perhaps this limitation should be disclosed more clearly.
 - Typo on line 99
 - Are there any missing relevant references for Proposition 1? My impression is that this result (or perhaps a small variation of it) is already known, but there are no references in Section 2.3. Indeed the proof of Proposition 1 says the argument is "standard", but again there are no references.
 - Is the hypothesized support set In Section 3 ordered? That is, is the definition of M-hat on line 183 allowed to re-order the x(1), ..., x(m)? It seems like this could greatly affect the value of M-hat, and line 177 (and common sense) suggests the set must indeed be ordered. However, sets are generally understood to be unordered, so the notation is a bit confusing.
 - Perhaps it could be explicitly emphasized that M-hat can become infinite large if the distribution of X is not sufficiently discrete, making the upper bound trivial.
 - On line 183, it should read 1-gamma instead of gamma.
 - In (5), perhaps one could recall that the denominator (alpha-delta-gamma) must be positive, as gamma may be large.
 -

**Time Spent Reviewing:**

4

---

> ### Author Response · Authors · 2021-08-10
> **Response to Reviewer tn9d**
>
> Thank you for your positive feedback and your many thoughtful suggestions! We will reply to individual comments below.
>
> 1. You commented "A possible limitation is the dubious practical relevance of this work because the proposed algorithm will only really be useful in cases where the explanatory variables can only take a small number of possible values compared to the sample size. In particular, this rules out all high-dimensional applications and all low-dimensional applications with continuous-valued variables”
> $ $
>
>     We agree that the proposed algorithm has some limitations - the focus of section 3 is to show that it is possible to achieve a bound that matches the lower bound in Theorem 1, especially in the regime $M < n^2$. As you mentioned, in most high-dimensional settings and settings with continuous variables where the support size is large($M > n^2$, to be specific), it would be hard to do a meaningful distribution-free inference - as Theorem 1 says. Our main result is that there are still some nontrivial cases where distribution-free inference is possible, and it would be interesting to explore how to build a practical inference procedure in those cases.
> $ $
>
>     In particular, the main aim of this work is to answer the question, “when is inference possible?”. For many related works in the distribution-free literature, the answer to this question is entirely negative (e.g., impossibility results for providing predictive inference that is valid pointwise at each $X$, for example). In the setting that we study here, it turns out that the answer is somewhat mixed -- inference is possible in certain regimes and not in others. We do not view the negative results as a drawback to our work; indeed, our goal is to accurately characterize the regimes in which distribution-free inference is a meaningful target, and thus positive and negative results in different regimes are to be expected.
> $ $
>
> 2. You commented "What happens if the distribution of $X$ is discrete but the dimensions d increase? I expect the upper bound should become trivial, as this problem seems very hard in high dimensions, and the algorithm relies on observing several identical values of $X$, which we know is extremely unlikely to occur in high dimensions. Perhaps a simple analytical example of this would be informative at the end of Section 3.”
> $ $
>
>     Our results say that what determines the possibility of distribution-free inference is the effective support size - so technically there’s no direct connection between the dimension and the possibility of inference. In the case of real examples, we may be likely to have an extremely large support in high dimensional settings, meaning that it would be hard to construct a meaningful distribution-free confidence interval(according to Theorem 1). On the other hand, we might also find some real data settings where the different covariates are highly redundant, and so even though the space of possible $X$ values is large (and grows with dimension), it might be the case that only a tiny fraction of this space is needed to capture most values that are actually observed in real data (and perhaps this would grow only very slowly with dimension).  We appreciate this thoughtful point and will elaborate more on this point in our next draft, and will attempt to find real data examples to illustrate the different types of behavior of the effective support size that we might see in practice.
> $ $
>
> 3. You commented “I find the title to be not very informative, and perhaps even a little confusing. In my opinion is does not convey clearly the idea the this work is about the construction of model-free confidence intervals for the conditional mean (model-free inference is typically predictive) and it does not suggest that it focuses on discrete explanatory variables (instead, I first thought the paper would be about classification).”
> $ $
>
>     Thank you for this helpful perspective. We will consider alternatives for the title.
> $ $
>
> 4. You commented “If this work has any concrete practical relevance it should be emphasized, perhaps by providing some examples of how it could be useful, and some comparisons of its performance against that of existing alternatives (possibly parametric ones, if non-parametric alternatives do not exist). If no meaningful applications come to mind, perhaps this limitation should be disclosed more clearly.”
> $ $
>
>     The proposed algorithm’s role in this paper is primarily to provide a theoretical verification of our lower bound (i.e., that our result in Theorem 1 is tight), so that we are confident that we have correctly understood the regimes in which inference is possible or impossible. We do not expect that this particular algorithm would necessarily be optimal in the possible regimes, but certainly believe that developing effective algorithms for these regimes is an important problem; our goal in this initial paper was to identify regimes where inference may be possible, but we certainly hope to explore potential algorithms more fully in future work, and will be more clear about these limitations in our next draft.
> $ $
>
> 5. You commented “Are there any missing relevant references for Proposition 1? My impression is that this result (or perhaps a small variation of it) is already known, but there are no references in Section 2.3. Indeed the proof of Proposition 1 says the argument is "standard", but again there are no references.”
> $ $
>
>     Thank you for this suggestion. We will do our best to find an appropriate reference for the next draft of our paper. However, our comment that the argument is “standard” was not intended to refer to this specific problem (i.e., distribution free confidence intervals for regression), but rather was meant more generally, as this type of construction comes up in many theoretical results (e.g., examining necessary assumptions for estimating a sample mean, in a non-regression context).
> $ $
>
> 6. You commented “Is the hypothesized support set In Section 3 ordered? That is, is the definition of M-hat on line 183 allowed to re-order the x(1), ..., x(m)? It seems like this could greatly affect the value of M-hat, and line 177 (and common sense) suggests the set must indeed be ordered. However, sets are generally understood to be unordered, so the notation is a bit confusing.”
> $ $
>
>     The set {$ x^{(1)} … x^{(m)} $} is indeed unordered in line 183, but the $x^{(m)}$’s are still ordered. To clarify, if we define the set
> $S_m =$  {$x^{(1)} … x^{(m)}$}, then our goal is to find the minimum m such that $S_m$ contains some minimum amount of the empirical probability. This means that the order of the $x^{(m)}$’s matters --- for example, $x^{(1)}$ is included in every $S_m$, while $x^{(2)}$ is included only in $S_m$ for $m\geq 2$, and so on. But inside of a single set $S_m$, the ordering of the m values in this set does not matter in order to answer the question, “how much empirical probability is captured by $S_m$”.
> $ $
>
> 7. You commented “Perhaps it could be explicitly emphasized that M-hat can become infinite large if the distribution of X is not sufficiently discrete, making the upper bound trivial.”
> $ $
>
>     Thank you for pointing this out, we agree that this needs to be clarified and we will do so in the next draft.
> $ $
>
> 8. You commented “On line 183, it should read 1-gamma instead of gamma.”
> $ $
>
>     Thank you, we will fix this error.
> $ $
>
> 9. You commented “In (5), perhaps one could recall that the denominator (alpha-delta-gamma) must be positive, as gamma may be large.”
> $ $
>
>     Thank you, we will make this clarification.

---

> > ### Comment · Reviewer_tn9d · 2021-08-26
> > **Re: author response**
> >
> > Thank you for taking my comments into consideration. I'm satisfied with these answers and I will keep my score unchanged.

---

### Official Review · Reviewer_uYq5 · 2021-07-14

**Rating:** 5
**Confidence:** 4

**Summary:**

The paper studies distribution-free prediction regions in regression problems, more precisely, it analyses confidence interval constructions, based on an i.i.d. data sample, for the true conditional expectation function (of the output given an input) at a new random input point, generated from the same distribution as the training inputs. Based on the concept of "effective sample size" it provides a lower bound for the expected length of this confidence interval (Theorem 1). The authors discuss some special cases, then provide an algorithm for discrete input spaces which, for certain "well-behaving" distributions (with "small" effective sample size), achieves a confidence interval length that matches the rate of the previous lower bound.

**Limitations And Societal Impact:**

The limitations are not adequately emphasized. Even the title suggests that the result is applicable for continuous distributions, but it is only partially true. Theorem 1 indeed applies to that case, as well, but in this respect it does not provide significantly better bounds than previous works. The real novelty of Theorem 1 is only for the discrete case, moreover, the results about the given algorithm (Theorem 3) only applies to the discrete case. Taking these into account, the title of the paper is misleading. The fact that the paper does not contain empirical evaluations is also a limitation.


**Main Review:**

The paper studies an interesting problem, i.e., distribution-free prediction regions, and tries to provide a positive result: a construction for vanishing-length prediction intervals. As the paper also discusses, such a construction is not possible without restricting the allowed distributions. The authors overcome this difficulty by restricting their attention to discrete input spaces and distributions for which the effective sample size is smaller than the square of the sample size. An over bound that can be applied even to continuous input spaces is also a valid and interesting result. The literature review part of the paper is average, it only discusses some better known results. Thought the assumptions needed for the results can be obtained from various parts of the paper, but it would have been much better to clearly highlight the assumption (e.g., by naming them, like A1, A2), starting from the one about an i.i.d. sample. The fact that the algorithm is limited to certain types of discrete distributions decreases the significance of the result (prediction regions are more interesting in the  continuous case). Finally, the paper does not contain any numerical experiments, though, it would have been interesting to see how the suggested method behaves in practice. Therefore, the results of the paper are promising (Theorem 1 and the algorithm achieving a matching upper bound), but they do not seem significant enough, and the quality of the presentation should also be improved (assumption, experiments, etc.).

--- --- ---

Post rebuttal comments: thank you for your answers. While it is certainly true that the definition of "effective support size" is well-defined for any distribution, it provides infinity for nonatomic ones. Therefore, the concept is not very useful to study continuous distributions. It is understandable that this paper has a theoretical focus, but the lack of numerical experiments also shows that the paper would require some extra work. Nevertheless, having read the other reviews, I think that the direction is promising and the ideas might be useful for researchers of this field, even in their current form. Hence, I have raised my rating.

**Time Spent Reviewing:**

5 hours

---

> ### Author Response · Authors · 2021-08-10
> **Response to Reviewer uYq5**
>
> Thank you for your thoughtful feedback and suggestions! Below we will reply to the individual points raised in your review.
>
> 1. You commented "The paper studies an interesting problem, i.e., distribution-free prediction regions, and tries to provide a positive result: a construction for vanishing-length prediction intervals. As the paper also discusses, such a construction is not possible without restricting the allowed distributions. The authors overcome this difficulty by restricting their attention to discrete input spaces and distributions for which the effective sample size is smaller than the square of the sample size.”
> $ $
>
>     We’d like to clarify that the goal of this work is to investigate different regimes based on the possibility of meaningful distribution-free inference. Our results(Theorem 1 & 3) show that the regimes are characterized by ‘effective support size’ - specifically, the possibility of having a vanishing-length distribution-free confidence interval depends on whether the effective support size is larger than $n^2$. However, it is not the case that we “restrict to discrete input spaces”. Our results are universal in the sense that they apply to every distribution, with the specific outcome of the result being determined by $M_\gamma(P_X)$ which is well-defined for all distributions whether discrete or not.
> $ $
>
> 2. You commented “The literature review part of the paper is average, it only discusses some better known results.”
> $ $
>
>     We appreciate your feedback and will try to expand our background and literature review in the next draft, as much as is possible given space constraints.
> $ $
>
> 3. You commented "The fact that the algorithm is limited to certain types of discrete distributions decreases the significance of the result.”
> $ $
>
>     Please see our reply to your point #1 above.
> $ $
>
> 4. You commented "the paper does not contain any numerical experiments, though, it would have been interesting to see how the suggested method behaves in practice”
> $ $
>
>     The focus of this work is on investigating this possibility theoretically rather than finding the best procedure -- the algorithm that we propose is primarily intended as a proof technique (to verify that the lower bound is indeed the correct scaling). We agree that it would be interesting to see how the procedure performs on real data, but we expect that it should be possible to substantially improve the algorithm and look forward to exploring this question in future work as well.

---

### Official Review · Reviewer_dAue · 2021-07-16

**Rating:** 7
**Confidence:** 3

**Summary:**

The paper investigates instance-dependent confidence intervals (CIs) for regression problems where the (marginal) distribution over covariates P_X has an atomic support. In particular, the main contribution of the paper is a lower bound on the measure of a CI in terms of an "effective dimension" of P_X which is roughly understood as a cardinality of the smallest subset of the support that captures most of the mass. Thanks to the instance-dependent analysis, the paper discovers two interesting regimes, namely when the effective dimension >> n^2, CI is non-vanishing asymptotically; on the other hand, when effective dimension << n^2, one can have a vanishing CI. The bound also organically recovers the fact (recently shown in other works) that CI is non-vanishing asymptotically when P_X is non-atomic. Finally, the paper matches the lower bound by proposing an algorithm for computing CI given an estimator, which matches a lower bound up to an additive excess risk of an estimator.

**Ethical Concerns:**

None.

**Limitations And Societal Impact:**

None.

**Main Review:**

The paper is very clearly written and does a good job positioning itself in the previous literature. The nice feature of this work is that it fills the gap between the continuous supports (where CIs are-vanishing) and finite supports in an instance-dependent way. The proposed algorithm (albeit, for a completely theoretical purpose) is simple enough to be used by practitioners. On that note, I found that the paper perhaps could say a bit more about other methods for computing confidence intervals (even in they are tailored for other setting); how would their upper bounds compare to the one obtained in Theorem 3?

**Time Spent Reviewing:**

2

---

> ### Author Response · Authors · 2021-08-10
> **Response to Reviewer dAue**
>
> Thank you for your encouraging feedback on our paper, and for your suggestions!
> You commented that "I found that the paper perhaps could say a bit more about other methods for computing confidence intervals (even if they are tailored for other setting); how would their upper bounds compare to the one obtained in Theorem 3?"
> $ $
> Thank you for this suggestion. Our procedure was designed to leverage the repeated observations at the atoms of the distribution so that the length of the confidence interval matches the lower bound of Theorem 1, as the goal of our work is to investigate the different regimes in terms of the possibility of meaningful distribution-free inference. Practical applications and comparison with other methods would also be an interesting and important topic to explore. In particular, we can add a brief comparison to the results one might obtain under smoothness assumptions on the function $\mu(x)$ (such as in the nonparametric inference literature).

---

### Official Review · Reviewer_WAr3 · 2021-07-16

**Rating:** 7
**Confidence:** 2

**Summary:**

In this paper the authors provide theoretical results on the type of problems that can be accurately learned (from a confidence interval standpoint) depending on the properties of the data distribution.
This departs from the usual convention of assuming as little as possible on the distribution, by adding a fairly minimal hypothesis that still provides strong results in the form of a lower bound (and upper bound) on the size of the attainable confidence intervals.

**Limitations And Societal Impact:**

The authors adequately addressed the limitations and potential negative societal impact of their work.

**Main Review:**

As this is an entirely theoretical paper, the breadth of the possible remarks is limited given the format of NeurIPS.

It is a well-written paper and as far as I can tell the proofs and results are correct and use standard techniques. I am however given the time constraint not able to confidently say everything is correct.

The three theorems themselves are novel and interesting and provide valuable insights on what type of distribution can be predicted with high confidence.

I think this paper, even if limited in impact to the general ML community, provides valuable insights and is of high quality. The proofs in themselves are of interest to other researchers working on similar problems.

**Time Spent Reviewing:**

4

---

> ### Author Response · Authors · 2021-08-10
> **Response to Reviewer WAr3**
>
> Thank you for your positive feedback on our paper! We would like to make a minor clarification in reply to your comment, “This departs from the usual convention of assuming as little as possible on the distribution, by adding a fairly minimal hypothesis that still provides strong results in the form of a lower bound (and upper bound) on the size of the attainable confidence intervals.”
> $ $
>
> The validity of our results does not rely on any assumptions -- in particular, we do not assume a particular value of $M$, or assume that the value of $M$ is in any way known. On the other hand, the extent to which output of any distribution-free procedure (including the simple one proposed in our paper) is able to be meaningful, i.e. to give a narrow/precise confidence interval, *is* dependent on $M$. (This is analogous to the flavor of many results in hypothesis testing, where Type I error control is guaranteed, while power might be strong or might be trivial, depending on the nature of problem.)

---

> > ### Comment · Reviewer_WAr3 · 2021-08-30
> > **Thank you**
> >
> > I see.
> >
> > Thank you for the clarification.
> >
> > I personally think this is a good paper and do not agree with uYq5 that more experiments are needed for publications. While more experiments are, of course, always welcome, considering that the contributions are mainly theoretical and interesting in themselves it's hard to fault the authors for the relative lack of experiments, especially since the format of conferences would force the authors to sacrifice some readability on the theoretical sections.
> >
> > I agree with the remarks of uYq5 to improve presentation, however, even if I can see why the notion of effective support can be seen as unsatisfying I feel this type of technical assumption is unavoidable.
> >
> > I am keeping my current score but lowering my confidence.

---

### Official Review · Reviewer_Rqsq · 2021-07-23

**Rating:** 7
**Confidence:** 4

**Summary:**

This paper is concerned with the problem of distribution-free conditional mean inference.
Specifically, assuming an iid sample of size n from a distribution P of (X,Y) in R^d*[0,1], the aim is to construct a confidence interval at point x for the conditional mean of Y given X=x.
Unlike what is sometimes considered in predictive inference, the goal is to construct a confidence interval containing the conditional mean, rather than a predictive interval containing the response (the latter would necessarily contain the "noise" and thus be of non-vanishing width).
The confidence interval should be valid distribution-free: for every distribution P on R^d*[0,1], the interval should contain the conditional mean at a random test point X, with probability 1-alpha over the draw of both the sample and the test point (drawn from the same distribution).
In addition to being universally valid, the interval/region should be of small/vanishing measure for distributions for which this is possible.

The main results are the following.

First, if the distribution of X is continuous, then no vanishing confidence intervals is possible. A general lower bound quantifying this is provided (Theorem 1), scaling as M^{1/4}/n^{1/2}, where M is a suitable notion of support size of the distribution of X. This implies in particular that, if n^2 << M and the distribution is noisy, then no non-trivial inference is feasible.

Second, a construction is provided, which takes as input an assumed value (or guess) of the support of X and conditional mean function, and outputs a confidence region. It is shown in Theorem 2 that the confidence interval is always valid, even if the guessed support of X and regression function are wrong. Then, it is shown in Theorem 3 that, if the assumed support of X and regression are correct, and if the distribution of X is not too far from uniform, then the confidence region is of average length at most M^{1/4}/n^{1/2}.

**Limitations And Societal Impact:**

Yes

**Main Review:**

The problem of distribution-free conditional mean inference considered in this paper is a simple but (in my opinion) interesting one. The paper is also well-written.

At the same time, I find the main positive result (Theorem 3), where the 'solution' to this problem is provided, somewhat disappointing. Indeed, for the procedure to yield small confidence regions, the procedure requires that both the support and the regression function are known (up to a small error for the regression, which appears in the bound). This takes away the core of the estimation problem, and in principle there is no need for such a strong a priori knowledge, as both the most frequent points and the value of the regression function at those points can be estimated from the sample.

Instead, the following simple procedure should be optimally adaptive in this setting. Given the test point X, find the points i such that X_i = X, and, using a suitable concentration inequality for means of bounded variables (such as Bernstein), form a confidence interval for the conditional mean based on the mean of the associated Y_i's. With a little bit of care in the conditioning (as the number N of points s.t. X_i = X is random), this yields a distribution-free valid inference procedure. In addition, this procedure should be essentially optimal among distribution-free valid ones.

The performance of this method, namely the average length of the confidence intervals (even if those were built with Hoeffding's inequality), should be at most: letting X the test point and N(X) the number of i's st X_i=X, and p(x) = P(X=x) for x in R^d,

E [ (N(X) + 1)^{-1/2} ]
= sum_x p(x) E (N(x) + 1)^{-1/2}
< c sum_{x st p(x) > 1/n} p(x) 1/[ p(x) n ]^{-1/2} + P (x : p(x) < 1/n)
< c n^{-1/2} sum_{x st p(x) > 1/n} p(x)^{1/2} + P (x : p(x) < 1/n),

which should be optimal for this problem in terms of dependence on P_X (improvements are possible in terms of the conditional variance of Y given X based on Bernstein).

While this may be quite simple, I would certainly recommend acceptance for a clean and well-written paper doing this, together with a matching lower bound. In fact, this is what I assumed (reading the abstract and introduction) this submission was doing, and I was therefore leaning for a strong accept before I got to Theorem 3, but it now seems like a somewhat borderline case.
In any case, due to the above restrictions, the positive results of Sec 3 seem less interesting or satisfying than the lower bound of Sec 2, so I would suggest dropping Section 3, and possibly replacing it with the rest of the proof of the lower bound from the Supplementary Material.
Perhaps the lower bound alone could warrant acceptance, depending on the other reviewers' and the AE's opinion?

Below are additional comments:

- In Theorem 1, the lower bound depends on (the quantile of) the conditional variance of the response given the output, and on the number of points.
This lower bound could perhaps be complemented in the noiseless case (the response is a function of covariates), in the regime n^2 < M, showing that no nontrivial inference (vanishing confidence region) can be obtained in this case.
This is due to the fact that, with large probability, the "test" point will not be in the sample, and since the distribution is unknown, the function value at this point could be arbitrary at this point, thus any 'valid' confidence region will need to have at least constant measure.
In addition, it might perhaps be possible to state the lower bound in terms of the complexity measure given above (with the sum of the p(x)^{1/2} and the total mass of low-probability points).

- Line 183: "to capture gamma". It seems that "to capture a fraction 1-gamma..." (or something equivalent) is meant.

==
Update in light of the authors' reply:
Thank you very much for clarifying the point of view taken in this paper, where estimation is not considered as part of the inference task, but rather decoupled from it.

I indeed assumed that building an estimator (with small error) should be part of the procedure, as it is needed to obtain good confidence intervals (since the error of the "guess" for the regression function appears in the length of the confidence region). While I stand with this principle, I also understand the perspective adopted here, and agree that from this perspective, the upper-bound construction is non-vacuous. I am therefore now in favor of acceptance, and will increase my score accordingly.

Incidentally, both my reservation about the upper bound, and my suggestion of the lower bound (first two points) are related to the fact that (unlike in your approach) I considered estimation of the regression function as part of the task. Indeed, in the regime sqrt(M) < n < M, it is not possible to estimate the regression function without additional structural assumptions (one cannot say anything about the regression function at x if x is not in the sample), so the error term in the bound will account for this effect. Admittedly, if for some reason one had a good estimator/guess (for some specific distribution), then indeed inference would be possible in this regime. So from one perspective, inference is impossible is the M>n regime (since even estimation is impossible), but from another, it is possible (a good estimator--even if it does not exist uniformly over this class--could be converted to a confidence band).

Overall, I appreciate that the point of view adopted here, while not fully satisfying in some respects, can also be defended from a certain perspective. In this spirit, I believe that the paper could significantly benefit from some additional explanation in the main text, for instance in the spirit of your reply. Indeed, as written, the algorithm assumes a (seemingly data-dependent?) guess for the true regression function and support. Adding some explanation (alluding to a sample-splitting approach, where a first part of the sample would be used to build an estimator) could be helpful. Perhaps more importantly, I would recommend adding some caveats, to give a more comprehensive picture: e.g., obtaining small confidence regions requires a good estimator, which does not exist in general for M>n (but can exist in special cases).

One could still argue that analysis of the estimation aspect should also be part of a thorough treatment of the question, but this would involve a major revision and is perhaps best left out of this version.

**Time Spent Reviewing:**

10

---

> ### Author Response · Authors · 2021-08-10
> **Response to Reviewer Rqsq**
>
> Thank you for your helpful suggestions and feedback! Below we will reply to the individual points raised in your review.
>
> 1. You commented "Indeed, for the procedure to yield small confidence regions, the procedure requires that both the support and the regression function are known (up to a small error for the regression, which appears in the bound). This takes away the core of the estimation problem, and in principle there is no need for such a strong a priori knowledge, as both the most frequent points and the value of the regression function at those points can be estimated from the sample …. Instead, the following simple procedure should be optimally adaptive in this setting...''
>    $ $
>
>     It is indeed true that our work does not consider the core estimation problem -- this is intentional, as it follows the spirit of much of the inference literature where the goal is to provide “wrapper” methods which can provide inference or uncertainty quantification for pre-existing algorithms; that is, inference methods that can be paired with any estimation algorithm that the analyst chooses to use. This type of approach appears in the conformal prediction literature for distribution-free prediction (which is a large part of the inspiration behind our work) where valid prediction intervals are built around the predictions coming from any “black box” prediction algorithm; as well as in more classical methods (for example, many multiple testing methods, such as the Benjamini-Hochberg procedure, provide a method to perform multiple testing given pre-computed p-values, which means that the analyst is free to produce these p-values using any desired algorithm). This decoupling of the two stages of the method, is part of the appeal of these “wrapper” methods, since they allow for valid inference to be built around any desired initial algorithm, meaning that we can use the most powerful algorithms at our disposal --- in other words, we do not have to limit ourselves to some specific, easy-to-analyze algorithm for the sake of enabling inference to be performed.
>    $ $
>
>     Regarding the specific procedure you suggest, this method will certainly be effective when the support size $M$ is small (much smaller than the training size $n$) so that each of the possible $X$ values is likely to appear multiple times in the training data. However, the main focus of section 3 is the nontrivial 'in-between' case where $n < M < n^2$ and the new observation $X$ is unlikely to have a value we've seen in the training data. This means that at a “typical” X value, we will have zero observations; even among the set of X values that we observe in our data set, at most of these values we will only have a single observation -- and so the proposed method would not be able to give meaningful intervals in this regime. The procedure that we propose in our paper is designed to be more universal in the sense that it can give nontrivial intervals even in the regime where $M > n$, as long as $M < n^2$ (i.e., matching the lower bound of Theorem 1).
>    $ $
>
> 2. You commented "In Theorem 1, the lower bound depends on (the quantile of) the conditional variance of the response given the output, and on the number of points. This lower bound could perhaps be complemented in the noiseless case (the response is a function of covariates), in the regime $n^2 < M$, showing that no nontrivial inference (vanishing confidence region) can be obtained in this case. This is due to the fact that, with large probability, the "test" point will not be in the sample..."
>    $ $
>
>     The lower bound in Theorem 1 was indeed obtained from an intuition similar to what you mentioned. One clarification that might be needed is that the $M > n^2$ regime comes from "how likely we have repeated observations in the training data" rather than "how likely our new $X$ has a value we've observed in the training data". The latter corresponds to the $M > n$ regime. The nontriviality of our result for the 'in-between' case($n < M < n^2$) basically stems from these interpretations: our result says that even if our new observation is likely to have a value that's not one of our previous observations, it is not impossible to do a meaningful distribution-free inference as long as there are some repeated values in the training data. To us, this result was initially counterintuitive -- how can we perform inference, with no assumptions, if the test X value was never seen? -- and the achievability of nontrivial inference in this regime is perhaps the most interesting outcome of this work, in our view.
>    $ $
>
> 3. You commented "Line 183: "to capture gamma". It seems that "to capture a fraction 1-gamma..." (or something equivalent) is meant."
>    $ $
>
>     Thank you for catching this error, we will be sure to fix it.

---

> > ### Comment · Reviewer_Rqsq · 2021-08-10
> > **Update**
> >
> > Thank you very much for clarifying the point of view taken in this paper, where estimation is not considered as part of the inference task, but rather decoupled from it.
> >
> > I indeed assumed that building an estimator (with small error) should be part of the procedure, as it is needed to obtain good confidence intervals (since the error of the "guess" for the regression function appears in the length of the confidence region).
> > While I stand with this principle, I also understand the perspective adopted here, and agree that from this perspective, the upper-bound construction is non-vacuous. I am therefore now in favor of acceptance, and will increase my score accordingly.
> >
> > Incidentally, both my reservation about the upper bound, and my suggestion of the lower bound (first two points) are related to the fact that (unlike in your approach) I considered estimation of the regression function as part of the task.
> > Indeed, in the regime sqrt(M) < n < M, it is not possible to estimate the regression function without additional structural assumptions (one cannot say anything about the regression function at x if x is not in the sample), so the error term in the bound will account for this effect. Admittedly, if for some reason one had a good estimator/guess (for some specific distribution), then indeed inference would be possible in this regime. So from one perspective, inference is impossible is the M>n regime (since even estimation is impossible), but from another, it is possible (a good estimator--even if it does not exist *uniformly* over this class--could be converted to a confidence band).
> >
> > Overall, I appreciate that the point of view adopted here, while not fully satisfying in some respects, can also be defended from a certain perspective. In this spirit, I believe that the paper could significantly benefit from some additional explanation in the main text, for instance in the spirit of your reply. Indeed, as written, the algorithm assumes a (seemingly data-dependent?) guess for the true regression function and support. Adding some explanation (alluding to a sample-splitting approach, where a first part of the sample would be used to build an estimator) could be helpful. Perhaps more importantly, I would recommend adding some caveats, to give a more comprehensive picture: e.g., obtaining small confidence regions requires a good estimator, which does not exist in general for M>n (but can exist in special cases).
> >
> > One could still argue that analysis of the estimation aspect should also be part of a thorough treatment of the question, but this would involve a major revision and is perhaps best left out of this version.

---

> > > ### Author Response · Authors · 2021-08-19
> > > **Thank you & a few replies**
> > >
> > > Thank you very much for your thoughtful comments!
> > > We have thought some more about this question of decoupling estimation and inference, and would like to add a few thoughts to continue the discussion.
> > >
> > > First, we would like to mention that many classical methods for providing inference / quantifying uncertainty / etc are very commonly presented in a decoupled way. For example, cross-validation is not typically taught as a second step that follows a specific estimation method; it is understood to be a tool that may be applied to any chosen model. Furthermore, since the primary goal of our paper is to study the lower bound (and to present the upper bound simply as a method for validating the lower bound), the power of the lower bound relies strongly on being independent of the estimation method (i.e., the standard minimax framework, where the bound holds uniformly over any estimation method), and so we feel it is natural to avoid too much emphasis on the estimation method in this particular work.
> > >
> > > Second, for regimes where M > n (or even where X is nonatomic), of course we agree that "it is not possible to estimate the regression function without additional structural assumptions", but we believe that structural assumptions are very natural in nearly any application -- the success of methods linear / parametric / smoothing-based / sparsity-based / etc would not be possible otherwise. Our perspective on the goals of distribution-free inference is not to avoid these (often useful and successful) assumptions, but instead to try to add "safeguards" by pairing them with inference whose reliability doesn't depend on the assumption being correct. So, if sqrt(M) < n < M, in this setting we would expect that for a "typical" problem, a reasonably chosen mu would indeed lead to nontrivial distribution-free intervals.

---

### Author Response · Authors · 2021-08-10
**General comment to the reviewers**

We thank the reviewers for their thoughtful feedback and suggestions on our paper. We have replied to each reviewer’s points individually in the comments.

---

### Decision · Program_Chairs · 2021-09-28

**Decision:**

Accept (Poster)

**Comment:**

The consensus of the reviewing committee is that the nice theoretical contributions of this paper are sufficient for a Neurips publication. There are some concerns about the importance of these results in applications and the authors are encouraged to address this concern in their revision.

**Consistency Experiment:**

NeurIPS has a long history of experimentation. In 2014, NeurIPS ran an experiment in which 10% of submissions were reviewed by two independent committees to quantify the randomness in the review process. This year, we repeated a variant of this experiment to see how the quality of the review process has changed over time.  This paper was part of the experiment and was therefore assigned to two committees (consisting of reviewers, an Area Chair, and a Senior Area Chair) that reached independent decisions.  If both committees made the same recommendation, this recommendation was followed. If a single committee recommended acceptance, the paper was accepted (with the exception of a few cases in which the other committee identified what we considered a fatal flaw, e.g., an error in a key result).

This copy’s committee reached the following decision: **Accept (Poster)**

The other committee assigned to the paper recommended **Reject**.  You can find the other set of reviews, along with any follow up discussion with the authors here:
https://openreview.net/forum?id=yuCiAWddUFq